# Complex system modeling reveals oxalate homeostasis is driven by diverse oxalate-degrading bacteria

Sromona D Mukherjee[1], Carlos Batagello[2], Ava Adler[3], Jose Agudelo[1], Anna Zampini[3], Mangesh Suryavanshi[1], Andrew Nguyen[4], Terry Orr[5], Denise Dearing[6], Manoj Monga[7], Aaron W Miller[1,3]*

[1]Department of Cardiovascular and Metabolic Sciences, Cleveland Clinic, Cleveland, United States; [2]Division of Urology, Hospital das Clínicas, University of Sao Paulo Medical School, Sao Paulo, Brazil; [3]Department of Urology, Glickman Urological and Kidney Institute, Cleveland Clinic, Cleveland, United States; [4]M Health Fairview Southdale Hospital, Edina, United States; [5]Department of Biology, New Mexico State University, Las Cruces, United States; [6]School of Biological Sciences, University of Utah, Salt Lake City, United States; [7]Department of Urology, University of California San Diego, San Diego, United States

*For correspondence:
millera25@ccf.org

Competing interest: The authors declare that no competing interests exist.

## eLife Assessment

This work presents a **valuable** approach based on a complex systems theoretical framework to characterize diet-host-microbe interactions and develop targeted bacteriotherapies through a three-phase workflow. Despite the partial support of the description and experimental setup of the 'complex systems theoretical approach,' the collected data are **solid** and advance our understanding of oxalate bacterial metabolism in microbial communities. This study will interest researchers working on gut microbiomes and the possible modulation of host-microbial interactions.

**Abstract** Decades of research have made clear that host-associated microbiomes touch all facets of health. However, effective therapies that target the microbiome have been elusive given its inherent complexity. Here, we experimentally examined diet-microbe-host interactions through a complex systems framework, centered on dietary oxalate. Using multiple, independent molecular, rodent, and in vitro experimental models, we found that microbiome composition influenced multiple oxalate-microbe-host interfaces. Importantly, the administration of the oxalate-degrading specialist, *Oxalobacter formigenes,* was only effective against a poor oxalate-degrading microbiota background and gives critical new insights into why clinical intervention trials with this species exhibit variable outcomes. Data suggest that, while heterogeneity in the microbiome impacts multiple diet-host-microbe interfaces, metabolic redundancy among diverse microorganisms in specific diet-microbe axes is a critical variable that may impact the efficacy of bacteriotherapies, which can help guide patient and probiotic selection criteria in probiotic clinical trials.

## Introduction

Research into the host-associated microbiome is at an inflection point. Decades of research have revealed that host-associated microbiomes are intimately linked to host health and touch all aspects of host physiology (*Gomaa, 2020*; *Cox et al., 2022*). The first phase of microbiome research was

focused on descriptive studies that characterized differences in the microbiome by host species, body site, and disease phenotypes (*Turnbaugh et al., 2007*; *Lloyd-Price et al., 2017*; *Wang and Jia, 2016*). Clinical case:control metagenome-wide association studies that found differences in microbiome composition between healthy and disease cohorts gave rise to the ambiguous term 'dysbiosis' and much of the literature is focused on 'balancing' the gut microbiota (*Tang and Hazen, 2024*). In the second, and current, phase of research, studies have moved towards gaining more mechanistic insight into specific microbe-host interactions that influence host physiology and disease (*Tang and Hazen, 2024*; *de Vos and de Vos, 2012*; *de Vos et al., 2022*; *Zimmermann et al., 2021*). However, it is increasingly being recognized that, given the wide variability in microbiome composition, host genetics, and lifestyles, understanding mechanisms and causal relationships between gut bacteria and host physiological responses is not enough for the development of effective bacteriotherapies that target the gut microbiota (*Vujkovic-Cvijin et al., 2020*; *Walter et al., 2020*). In fact, many probiotic clinical trials, with known mechanistic links between the probiotics and host physiology, have exhibited wide variability in results (*Barkhidarian et al., 2021*; *Rogers and Mousa, 2012*; *Ceccherini et al., 2022*; *Batagello et al., 2018*). It is currently unclear what sources of variability may drive the response to targeted bacteriotherapies. Some variables that have been proposed include heterogeneity in microbiome composition, alcohol consumption, and bowel movement quality (*Vujkovic-Cvijin et al., 2020*).

Complex system theory is one means to potentially constrain hypotheses and bridge the gap between mechanistic studies and effective bacteriotherapies (*Foster et al., 2008*; *Layeghifard et al., 2017*). Studies of the host-associated microbiome often casually acknowledge that these systems are complex and, while, there has been progress made in developing mathematical approaches to complex modeling of host-microbiome relationships (*Foster et al., 2008*; *Layeghifard et al., 2017*; *Ridenhour et al., 2017*; *Kumar et al., 2019*), few studies have bridged the gap between experimental biology and complex system theory.

Mammals and their gut microbiome, are often considered as a holobiont (*Richardson, 2017*), defined as a discrete unit that exhibits collective action and evolves as a unit. Using evolutionary theory approaches, it is hypothesized that the form and function of the microbial part of the holobiont is driven primarily by host-microbe influences, but which require microbe-microbe interactions to help manage the enormous burden of microbiome constraint, which is termed an 'ecosystem on a leash' (*Foster et al., 2017*). Under this hypothesis, microbe-host interactions, in which host-associated microbes evolve to produce metabolic by-products for the sole benefit of the host, are necessarily rare. Essential features of complex systems include functional redundancy (*Foster et al., 2017*; *Palla et al., 2005*) and cooperation through chains of direct interactions (*Foster et al., 2017*; *Palla et al., 2005*; *Lambiotte et al., 2019*), distinct functional nodes that process and transfer resources (*Palla et al., 2005*; *Lambiotte et al., 2019*), fractality (*Palla et al., 2005*; *Lambiotte et al., 2019*), and adaptability through specific homeostatic feedback mechanisms to maintain relatively consistent internal conditions given a dynamic external environment (*Foster et al., 2017*; *Cohen et al., 2022*). While considerable progress has been made in understanding the nature of complex systems, there is a lack of consensus on basic terminology and broadly applicable analytical or experimental models (*Torres et al., 2021*). This limitation is due in part because complex systems necessarily transect many divergent entities in nature and is thus studied in diverse scientific disciplines (*Torres et al., 2021*). Thus, while some essential features of complexity have been elucidated within specific real-world systems, such as the conversion of polysaccharides to butyrate by multiple species in the gut (*Roediger, 1980*), or feedback loops in blood pressure regulation (*Javorka et al., 2017*), experimental evidence of multiple features of complexity within a single system is sparse, especially in microbe-host systems.

To overcome the limitations of applying complex systems theory to the microbe-host holobiont, new models and experimental frameworks are needed that balance complexity with tractability. Antinutrients, molecules produced in plants to deter herbivory, disrupt homeostasis by targeting the function of the microbiome, host, or both (*Kolodziejczyk et al., 2019*), and provide an effective focal point to study complexity. Oxalate is an antinutrient present in many plant-based foods, which typically provides the majority of oxalate in circulation (*Crivelli et al., 2020*), but is also produced as a terminal metabolite in the liver (*Holmes et al., 2001*). While some host genetic mutations increase endogenous production of oxalate (*Cochat and Rumsby, 2013*), mammals do not produce enzymes capable of degrading oxalate (*Hodgkinson, 1977*). However,

multiple oxalate-degrading bacteria exist in the gut, which degrades oxalate through one of a handful of simple metabolic pathways involving one or two genes (*Allison et al., 1985*; *Miller and Dearing, 2013*; *Liu et al., 2021a*), which isolates this function to the gut microbiota. As such, oxalate degradation as a function, exhibits a moderate amount of complexity compared to other gut microbiota functions, such as the production of trimethylamine N-oxide (TMAO), that requires host input (*Tang et al., 2015*). Beyond the moderate complexity, it is known that oxalate-degrading bacteria are susceptible to antibiotics and that antibiotic use decreases oxalate degradation (*Nazzal et al., 2021*; *Miller et al., 2019b*; *Kharlamb et al., 2011*; *Lange et al., 2012*; *Mittal et al., 2005*). Elevated levels of oxalate induce oxidative stress, activate the inflammasome, and disrupt epithelial barrier function through tight junction proteins (*Mulay et al., 2013*; *Peerapen and Thongboonkerd, 2011*; *Peerapen and Thongboonkerd, 2021*). This molecule has been linked to diseases including kidney stones and chronic kidney disease (*Holmes et al., 2016*; *Nazzal et al., 2016*; *Waikar et al., 2019*), breast cancer (*Castellaro et al., 2015*), and cardiometabolic disorders such as atherosclerosis (*Liu et al., 2021b*), obesity (*Efe et al., 2019*), and diabetes (*Efe et al., 2019*). *Oxalobacter formigenes,* which uses oxalate as a sole carbon and energy source (*Allison et al., 1985*), is a well-studied oxalate-degrading species of bacteria. The negative, at times lethal, effects of oxalate, along with the identification and mechanisms of oxalate-degrading bacteria in the gut, have been worked out for several decades (*Allison et al., 1985*; *James and Butcher, 1972*; *Anantharam et al., 1989*; *Baetz and Allison, 1989*; *Baetz and Allison, 1990*), pre-dating the current microbiome era.

Despite this knowledge, clinical intervention trials involving *Oxalobacter formigenes,* which is perhaps the most effective oxalate-degrading species known, have successfully resulted in a significant reduction in urine oxalate levels in only 43% of studies (*Batagello et al., 2018*; *Ariceta et al., 2023*). Oxalate-degrading lactic acid bacteria have been successful in 37.5% of studies (*Batagello et al., 2018*). Both treatments led to a wide variability in patient responses. Given these data and the well worked out mechanisms of oxalate metabolism, it is clear that having an understanding of the mechanistic links between the gut microbiota and host physiology, alone, is demonstrably not sufficient to develop effective bacteriotherapies. Therefore, even though the history of oxalate-microbe-host interactions is much greater than most other microbe-host systems, oxalate degradation represents an accurate reflection of the challenges faced by microbiome research and is a prime candidate for complex system modelling to understand the critical variables that determine responsiveness to bacteriotherapies.

The objectives of the current study were to evaluate oxalate-microbe-host interactions, within the framework of complex systems. We used multi-omic approaches utilizing multiple independent in vivo and in vitro models to understand the critical variables that influence the gut microbiota's maintenance of oxalate homeostasis and its impact on the host. We targeted several potential oxalate-microbe-host interfaces that include the gut microbiota itself, which is one of the first lines of defense against antinutrients (*Dearing and Weinstein, 2022*; *Kohl et al., 2014b*), intestinal epithelium, which is an important barrier between microbe and host (*Foster et al., 2017*), the liver, which is important for the biotransformation of dietary or microbial metabolites (*Grant, 1991*), along with the kidneys and vasculature, where calcium oxalate can potentially form calcified deposits or induce inflammation (*Liu et al., 2021b*; *Robertson and Peacock, 1980*; *Nishizawa et al., 2023*; *Fishbein et al., 2008*). Collectively, data indicate that multiple oxalate-microbe-host interfaces are influenced significantly by gut microbiota composition and that harboring diverse microorganisms capable of degrading oxalate can limit the impact of *O. formigenes* as a probiotic. Based on results from this study, we propose a phased approach for the development of bacteriotherapies whereby clinical case:control studies determine whether or not a clinical phenotype is associated with the microbiome and the microbial taxa/functions associated with specific phenotypes (Phase I).

Hypothesized taxa and/or microbial functions should be mechanistically resolved through in vitro and germ-free animal studies (Phase II). Finally, mechanistic insights should be applied to a complex systems theoretical framework to identify those variables that most impact the potential success of bacteriotherapies (Phase III). Such a phased approach would have broad implications for patient and probiotic selection in the development of targeted bacteriotherapies.

## Results

### Defining the oxalate-microbe-host as a complex system

To apply complex systems modeling to oxalate-microbe-host interactions, we first defined the system as a network of nodes, connections, and fractal layers. Nodes represent distinct sites where oxalate is processed, connections define the transfer of oxalate or its metabolites between nodes, and fractals represent autonomous subunits that perform analogous but distinct functions. For antinutrients like oxalate, we identified four primary nodes where transformation occurs or where oxalate directly affects the host. The first node is the stomach, where acidic conditions can chemically modify some molecules, such as glucosinolates (*Frandsen et al., 2019*). A fraction of oxalate may be absorbed into circulation at this stage.

The second node is the intestine, where oxalate encounters high bacterial densities, particularly in the colon. Within the gut microbiome, two fractal layers exist: microbial species and the genes within each species. Microbial oxalate degradation follows one of three possible models: (a) one species/ one function, (b) multiple species/one function, or (c) multiple species/multiple functions. Historically, *Oxalobacter formigenes* was considered the primary oxalate degrader (*Daniel et al., 2021*), relying on a two-gene metabolic pathway (oxalyl-CoA decarboxylase and formyl-CoA transferase) (*Baetz and Allison, 1989*; *Baetz and Allison, 1990*). However, more recent studies demonstrate that oxalate degradation is a broader function distributed across multiple bacterial species, utilizing diverse metabolic pathways that may require one or two genes (*Miller and Dearing, 2013*; *Liu et al., 2021a*; *Turroni et al., 2010*; *Turroni et al., 2007*). Additionally, gut microbiota interactions influence oxalate degradation (*Belenguer et al., 2013*; *Miller et al., 2019a*; *Miller et al., 2017a*; *Miller et al., 2017b*; *Miller et al., 2014*; *Miller et al., 2016b*; *Miller et al., 2016a*) with by-products such as $CO_2$ and formate serving as substrates for downstream metabolic pathways, including acetogenesis, methanogenesis, and sulfate reduction (*Drake, 2012*; *Asanuma et al., 1999*; *Ferry, 2012*; *Oremland and Polcin, 1982*). Given the inconsistent results of *O. formigenes* trials and the recognition of broader oxalate-microbiota interactions, we hypothesize that oxalate homeostasis is maintained either by diverse oxalate-degrading bacteria or by cooperative networks in which a subset of bacteria degrades oxalate while others utilize its metabolic by-products.

A portion of intestinal oxalate is absorbed into circulation or excreted in the stool, while another fraction is secreted back into the gut via transporters such as SLC26A6 (*Freel et al., 2006*; *Hatch and Freel, 2008*). Additionally, some oxalate is transported via the portal vein to the liver, forming the third node (*Dancygier and Dancygier, 2010*).

Unlike other metabolites that are detoxified by the liver, oxalate is a source rather than a sink in hepatic metabolism (*Hodgkinson, 1977*). However, the liver plays a central role in metabolizing other gut-derived molecules, such as trimethylamine, which it converts into the more pathogenic TMAO (*Tang et al., 2015*). Some complex antinutrients, such as creosote, require both gut microbial and hepatic metabolism for complete degradation (*Kohl and Dearing, 2011*). The liver itself can be divided into fractal layers based on cell type and gene expression. The final node includes systemic circulation and target organs, where unmetabolized oxalate or other antinutrients may trigger immune responses before eventual urinary excretion.

This framework generates multiple testable hypotheses regarding the cooperative roles of gut microbiota, hepatic metabolism, and host physiology in oxalate homeostasis, which are explored in subsequent sections.

### Constitutive and oxalate-dependent effects of the microbiome impact host hepatic gene expression

Both the gut microbiota and the liver are important organs for the neutralization of antinutrients (*Grant, 1991*; *Kohl and Dearing, 2016*), indicative of functional redundancy, cooperation, and fractality. While it is known that the liver does not degrade oxalate itself, it could still be impacted by oxalate, in gut microbiota-dependent ways. To determine the effects of oxalate exposure and microbiome on host hepatic gene expression, we used a fecal transplant model to examine different host microbiomes with the same host genetics. The microbiota from two different host species were utilized for these experiments. Specifically, Swiss-Webster mice harbor a poor oxalate-degrading microbiota and are often used in studies of hyperoxaluria (*Miller et al., 2019b*; *Miller et al., 2017b*; *Miller et al., 2016a*; *Hatch et al., 2006*; *Hatch and Freel, 2013*). Conversely, the white-throated

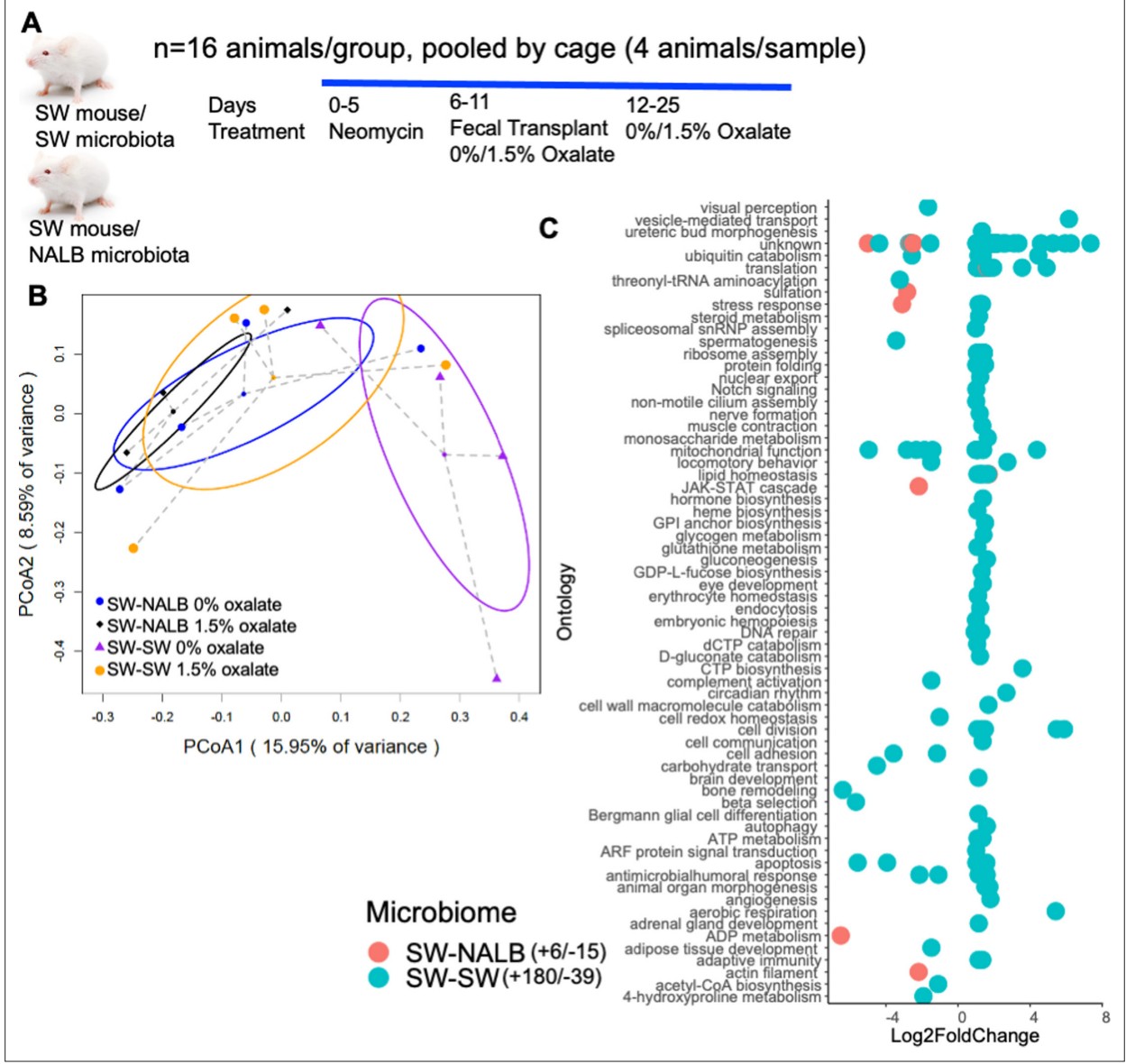

**Figure 1.** Oxalate exposure impacted host hepatic activity in a microbiota-dependent fashion. (**A**) Swiss Webster mice were given neomycin, followed by an allograft (Swiss-Webster mice, SWM) or xenograft (*N. albigula*, NALB) fecal transplant, then maintained on a 0 or 1.5% oxalate diet prior to necropsy for fecal metabolomics (*Figure 2*) and hepatic transcriptomics. (**B**) PCoA of normalized, whole-transcriptome data. p=0.02 for microbiome composition, p=0.07 for dietary oxalate content, p=0.3 in two-way analysis; two-way PERMANOVA. (**C**) Total number of hepatic genes significantly stimulated or inhibited by dietary oxalate. Significant genes are plotted by Log2FoldChange. Positive values reflect genes increased with oxalate exposure and negative values are genes decreased with exposure. False discovery rate (FDR) <0.05, Wald test. Hepatic genes are annotated to pathway (Kegg, Uniprot, PubChem, Metacyc) and the total number of genes that exhibit a positive or negative shift with oxalate exposure are listed in the legend. Complete gene list is in *Supplementary file 2*.

woodrat, *Neotoma albigula,* is a wild rodent that exclusively consumes *Opuntia* cactus, which is high in oxalate, and harbors a highly effective and responsive gut microbiota tuned to oxalate degradation (*Miller et al., 2017a*; *Miller et al., 2014*; *Miller et al., 2016b*; *Justice, 1985*; *Kohl et al., 2014a*). For the transplant model, a 5-d course of neomycin was used to suppress the native gut microbiota of Swiss-Webster mice (SWM) (*Miller et al., 2019b*), followed by fecal transplants either from SWM (allograft; SW-SW) or *N. albigula* (NALB), which has a highly effective and transferable oxalate-degrading gut microbiota (*Miller et al., 2017a*; *Miller et al., 2017b*; *Miller et al., 2016a*), (xenograft; SW-NALB). Subsequently, animals were fed either a 0% or 1.5% oxalate diet (Evigo, *Figure 1A*, *Supplementary file 1*). Liver tissue was obtained after 3 wk and processed for bulk

RNAseq. The RNAseq data analyses revealed constitutive microbiome and microbiome-dependent oxalate effects (*Figure 1B and C*), whereby SW-SW mice exhibited an oxalate-dependent alteration of 219 hepatic genes, with a net increase in activity, while the SW-NALB mice exhibited an oxalate-dependent alteration of 21 genes with a net decrease in activity (*Figure 1C*, *Supplementary file 2*). In the SW-NALB mice, the primary response was a decrease in sulfation activity with oxalate exposure, which is involved in the deactivation, detoxication, and excretion of xenobiotics (*Figure 1C*), and suggests oxalate may be beneficial for this host-microbe system (*Koppel et al., 2017*). The primary response in the SW-SW mice was an increase in mitochondrial activity, translation, protein regulation, and ribosome biogenesis, indicative of oxalate-induced hepatic stress (*Liu et al., 2021b*; *Strzelecki and Menon, 1986*). Since hosts only differed by gut microbiota composition, these data demonstrate causative interactions between the gut microbiota and liver activity through oxalate-dependent and independent pathways. Within the framework of complex systems, results show microbe-host cooperation whereby oxalate effectively processed within the SW-NALB gut microbiota (*Miller et al., 2019b*; *Miller et al., 2017a*; *Miller et al., 2017b*; *Miller et al., 2016a*) reduced overall liver activity, indicative of a beneficial impact. Data also suggest that both the gut microbiota and the immune system are involved in oxalate remediation (redundancy), such that if oxalate cannot be neutralized in the gut microbiota or liver, then the molecule will be processed through host immune response mechanisms (fractality), in this case indicated through an overall increase in inflammatory cytokines with oxalate exposure combined with an ineffective oxalate-degrading microbiota (*Supplementary file 6a, b*; *Supplementary file 9a; b*).

## Constitutive and oxalate-dependent effects of the microbiome impact microbial metabolic activity in the gut

Microbe-microbe interactions are important features of the microbe-host holobiont within the context of complex systems, both in terms of constraining the microbiome and in processing dietary components *Foster et al., 2017*; *Palla et al., 2005*; *Roediger, 1980*; *Dearing and Weinstein, 2022*. To assess changes to microbial metabolic output with oxalate exposure, we used the same animals as above. Following the diet trial, colon stool was collected post-necropsy and processed for untargeted metabolomics, which is a measure of total metabolites present in stool from the diet, microbial activity, and host activity. Collectively, results are indicative of constitutive microbiome and microbiome-dependent oxalate effects (*Figure 2A and B*). The SW-SW mice exhibited an alteration of 162 microbial metabolites upon oxalate exposure, with a net decrease in activity compared to the no oxalate group, whereas the SW-NALB mice exhibited an alteration of 83 microbial metabolites, with a net increase in activity compared to the no oxalate group (*Figure 2B*, *Supplementary file 3*). In SW-NALB mice, the primary response was an increase in lipid metabolism and a shift (increase/decrease) in fatty acid, secondary metabolite, and alkaloid profiles. In the SW-SW mice, the primary response was an increase in fatty acid synthesis and phenylalanine metabolism, and a decrease in the synthesis of secondary metabolites, cholesterol, and alkaloids (*Figure 2B*). Therefore, while oxalate had a much greater impact on gut microbial metabolism in SW-SW mice overall, the metabolic pathways impacted were similar. Net changes in microbial metabolites produced are indicative of a negative impact of oxalate on microbial activity in SW-SW and a positive impact in SW-NALB mice. Integration of host hepatic gene expression and gut metabolomic data shows that oxalate induces a small decrease in hepatic activity overall for SW-NALB mice and a large increase in hepatic activity for SW-SW mice (*Figure 2C*, *Supplementary file 4*). From a complex systems perspective, data reflect a causative effect of oxalate for the shift in microbial metabolic output. Specifically, the SW-NALB mice exhibit hallmarks of homeostatic feedback with oxalate exposure to maintain a consistent metabolic output, defined by the relatively small, net negative, microbial metabolite-hepatic gene network compared to the large, net positive, network of SW-SW mice. Additionally, data further support the cooperation, redundancy, and fractality of the gut-liver axis. While SW-NALB exhibit a small increase in microbial metabolic activity and a decrease in liver activity, the SW-SW mice saw a large decrease in microbial metabolic activity, coupled with a large increase in hepatic activity, which is reflected in the multi-omic network profiles (*Figure 2C*) and suggests that the oxalate-induced change in microbial metabolism is responsible for the change in hepatic activity.

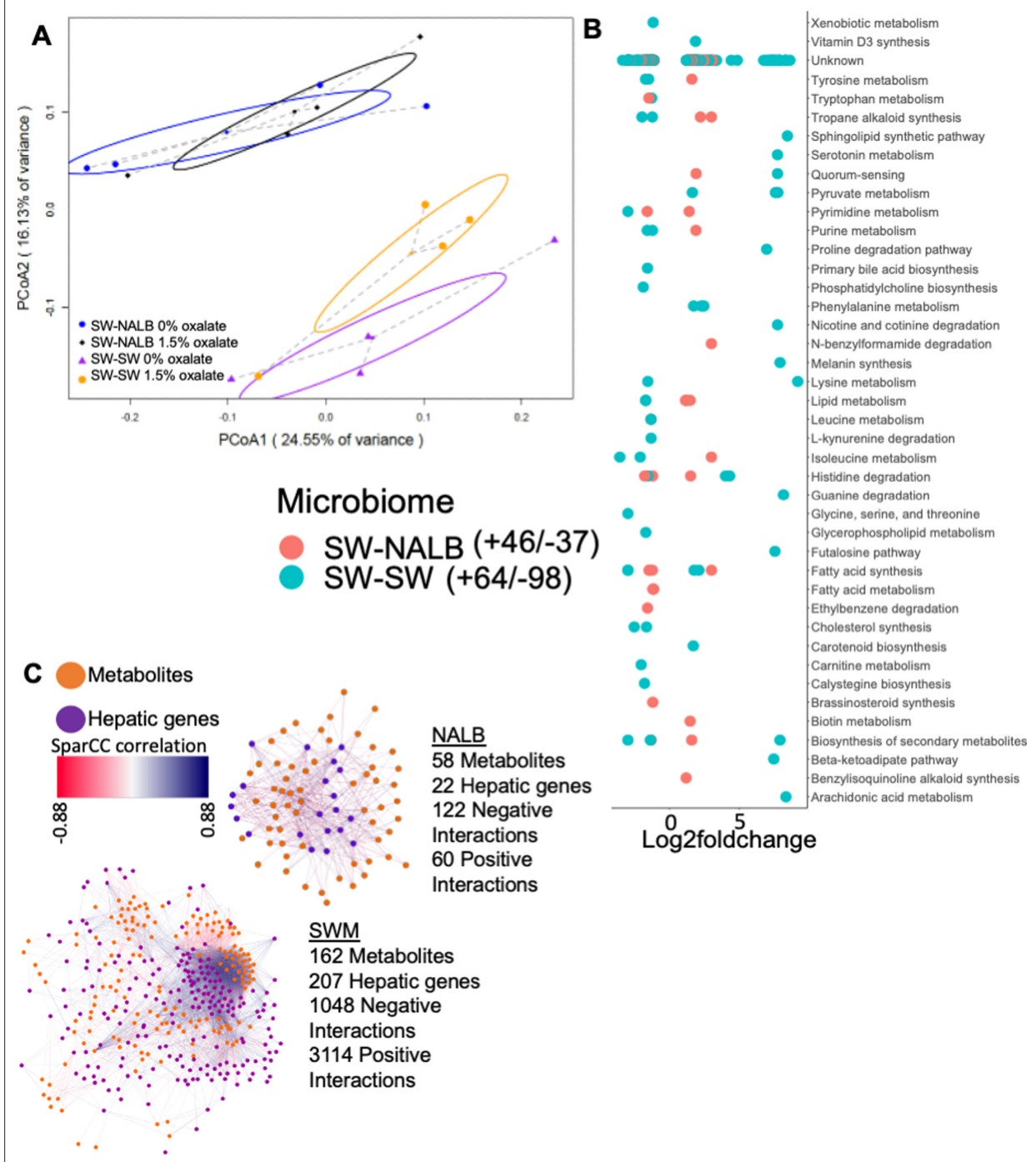

**Figure 2.** Oxalate exposure impacted microbial metabolic activity in a microbiota dependent fashion. (**A**) PCoA of protein normalized, log-transformed metabolomic data. p=0.001 for microbiome composition, p=0.1 for dietary oxalate content, p=0.3 in two-way analysis; two-way PERMANOVA. (**B**) Total number of fecal metabolites significantly stimulated or inhibited by dietary oxalate. Significant metabolites are plotted by Log2FoldChange. Positive values reflect metabolites increased with oxalate exposure and negative values are metabolites decreased with exposure. False discovery rate (FDR) <0.05, Mann-Whitney U or Fisher's exact test. Metabolites are annotated to pathway (Kegg, Uniprot, PubChem, Metacyc) the total number of metabolites that exhibit a positive or negative shift with oxalate exposure are listed in the legend. Complete list is in ***Supplementary file 3***. (**C**) Change in the host-microbe interaction network upon exposure to oxalate, quantified as hepatic gene-microbe metabolite correlations R >+/-0.3 and FDR <0.05 with SparCC, visualized in Cytoscape. All host-microbe interactions are listed in ***Supplementary file 4***.

## Oxalate stimulates the growth of microbial populations involved in oxalate metabolism, formate metabolism, and their precursors

To gain a deeper understanding of microbe-microbe interactions associated with oxalate exposure and to identify specific microorganisms that positively respond to oxalate exposure, NALB with their native microbiota were pair-fed increasing amounts of oxalate, from 0.2 to 6% (*Figure 3A*). Stool was sampled after 5 d on the 0.2% and 6% diets and processed for shotgun metagenomics. Analysis of metagenomic data revealed that oxalate had a significant impact on metagenome composition (*Figure 3B*), with a significant increase of 1073 gene populations and a decrease of 382 gene populations (*Figure 3C*; *Supplementary file 5*). Changes in gene abundance included four oxalate metabolism gene populations, a shift in 48 gene populations involved in formate metabolism (32+/16 -), and a shift in genes related to glycine and glyoxylate/dicarboxylate metabolism (6+/4-), which are precursors to oxalate (*Figure 3D*). A total of 128 differentially abundant genes were involved in sugar metabolism. Altered gene populations primarily belonged to *O. formigenes* and *Alistipes senegalensis*, with many genes belonging to the *Muribaculum* genus (*Figure 3E*). *Muribaculum spp* harbor oxalate-degrading genes, which suggests oxalate metabolic redundancy in the gut microbiota (*Ormerod et al., 2016*). The oxalate-dependent metagenomic divergence of the NALB gut microbiota (*Figure 3*), which has been replicated in multiple studies on the NALB gut microbiota (*Miller et al., 2017a*; *Miller et al., 2016b*), combined with the lack of change in the NALB microbial metabolomic profile with oxalate exposure (*Figure 2*, SW-NALB mice), suggest that oxalate stimulates taxonomically diverse, but metabolically redundant microorganisms, in support of maintaining homeostasis. Given that data came from the same hosts sampled longitudinally, these data also reflect a microbiota that is adaptive to oxalate exposure, which is another important characteristic of complex systems, and suggests that metabolic redundancy is critical for oxalate homeostasis.

## Oxalate and formate metabolism are highly redundant functions in the NALB gut

To investigate the hypothesis that oxalate stimulates a taxonomically diverse, metabolically redundant community, 248 full-length genomes were extracted from shotgun metagenomic data (*Figure 3—figure supplement 1*). Genes for oxalate metabolism, formate metabolism, or the formate metabolic pathways of acetogenesis, methanogenesis, and sulfate reduction, were derived from the KEGG pathway database (*Okuda et al., 2008*) and mapped to full-length genofmes. This analysis provides a targeted assessment of the potential for metabolic redundancy aimed at oxalate metabolism and pathways associated with the by-products of oxalate metabolism (potential cooperation). A total of 59.3% of genomes contained at least one gene associated with oxalate metabolism or handling (*Figure 4A*), most represented by oxalyl-CoA decarboxylase, glycerate dehydrogenase, and formyl-CoA:oxalate CoA transferase (*Figure 4B*). However, only 27.8% of genomes harbored a complete metabolic pathway for oxalate degradation (*Figure 4C*). Taxa with oxalate genes were dominated by *Bacteroides*, *Muribaculaceae*, *Clostridium*, *Ruminococcus*, and *Lachnospiraceae* (*Figure 4D*). Formate metabolism genes were found in 97.18% of genomes, which was dominated by serine hydroxymethyltransferase, and formate-tetrahydrofolate ligase (*Figure 3—figure supplement 2A–C*). Acetogenic genes were also present in 97.18% of genomes, dominated by acetate kinase and formate-tetrahydrofolate ligase (*Figure 3—figure supplement 3A–C*). Methanogenic genes were present in 100% of genomes, dominated by phosphoserine phosphatase, atp-dependent 6-phosphofructokinase, and phosphate acetyltransferase (*Figure 3—figure supplement 4A–C*). Sulfate-reducing genes were present in 31.05% of genomes, dominated by bifunctional oligoribonuclease and PAP phosphatase, FMN reductase, and cysteine synthase (*Figure 3—figure supplement 5A–C*). Data show highly redundant oxalate-associated metabolic pathways and thus provide evidence for very robust homeostatic feedback mechanisms to handle oxalate and metabolic by-products within the NALB gut microbiota. Additionally, the broad diversity of species that contain oxalate-related genes suggests that the distribution of metabolic genes is somewhat independent of the distribution of microbial species, which suggests that microbial genes exist in an autonomous fractal layer, to some degree. This hypothesis is supported by studies which show a high degree of horizontal gene transfer within the gut microbiota as a means of adaptation (*Groussin et al., 2021*).

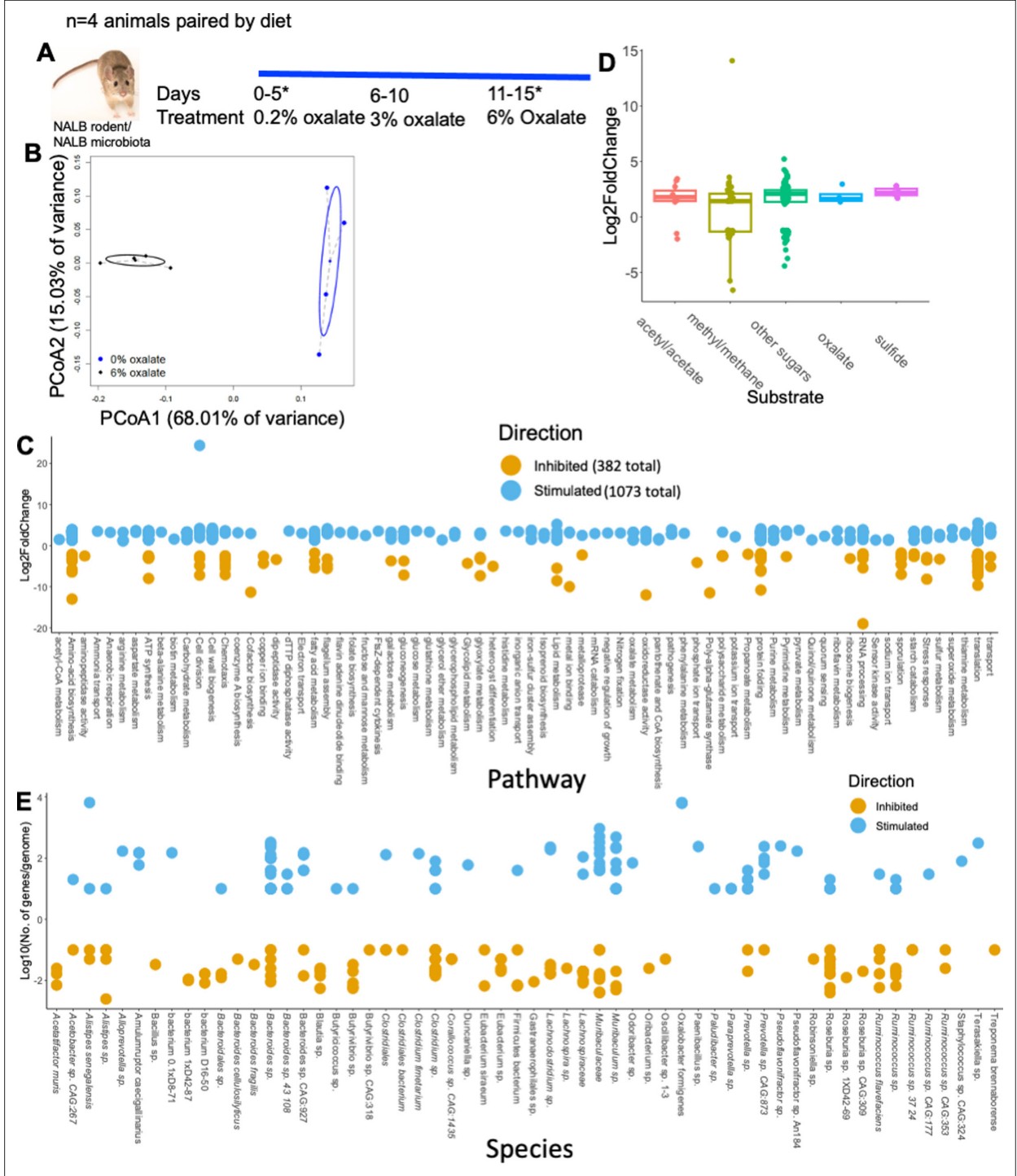

**Figure 3.** Oxalate exposure stimulates taxonomically diverse microorganisms, with a few strains that dominate the response. (**A**) *Neotoma albigula* with native microbiota were given increasing amounts of dietary oxalate up to 6% w/w. *indicates sampling timepoints. (**B**) PCoA of normalized metagenomic data. p=0.02; PERMANOVA. (**C**) Total number of microbial genes significantly stimulated or inhibited by dietary oxalate, annotated to pathway (Kegg, UniProt, PubChem, Metacyc) and listed by Log2FoldChange. The total number of genes stimulated or inhibited by oxalate are listed in the legend. Genes with unknown annotation are not listed. The complete list of annotated genes is listed in ***Supplementary file 5***. (**D**) Number of significantly differentiated genes involved in oxalate degradation, sulfate reduction, acetogenic, methanogenic, or sugar metabolic pathways or utilization of by-products of those pathways stimulated (positive) or inhibited (negative) by dietary oxalate. Genes are listed by their Log2FoldChange between no and high oxalate diets. False discovery rate (FDR) <0.05, Wald test. (**E**) The number of genes/genome significantly altered by oxalate, mapped to microbial

*Figure 3 continued on next page*

*Figure 3 continued*

genomes extracted from *N. albigula.* Number of genes/genomes are log10-transformed to show the distribution more clearly. A total of 92% of genomes had at least one significantly altered gene population mapped to them.

The online version of this article includes the following figure supplement(s) for figure 3:

**Figure supplement 1.** Phylogeny of genomes extracted from the *N. albigula* metagenome, in comparison to >3000 full-length microbial genomes.

**Figure supplement 2.** Representation of formate metabolism genes in the *N. albigula* (NALB) metagenome derived from 248 full-length genomes.

**Figure supplement 3.** Representation of acetogenic genes in the *N. albigula* (NALB) metagenome derived from 248 full-length genomes.

**Figure supplement 4.** Representation of methanogenic genes in the *N. albigula* (NALB) metagenome derived from 248 full-length genomes.

**Figure supplement 5.** Representation of sulfate-reducing genes in the *N. albigula* (NALB) metagenome derived from 248 full-length genomes.

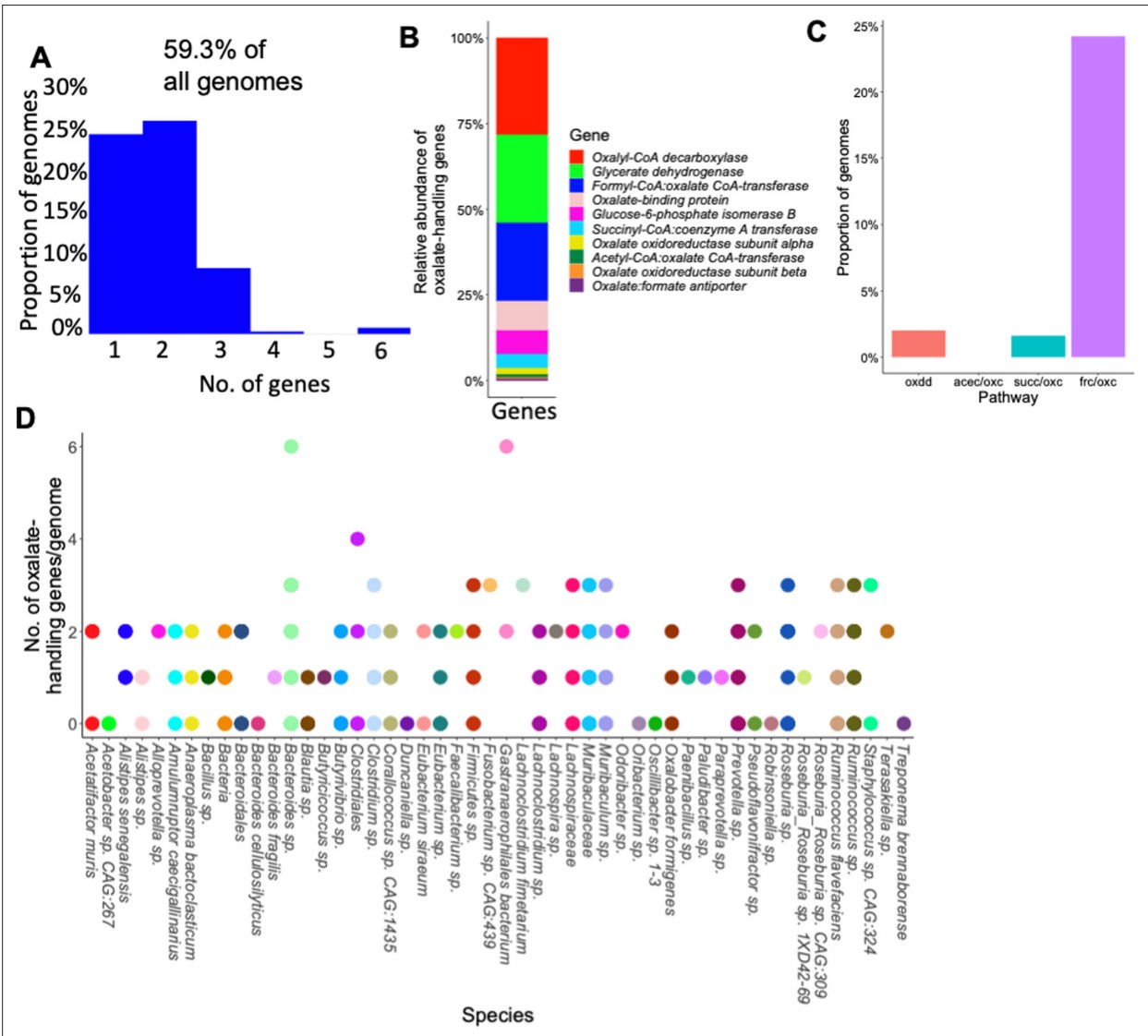

**Figure 4.** Genes related to the metabolism or handling of oxalate are present in >50% of 248 full-length *N. albigula* (NALB) microbial genomes from the gut. (**A**) Proportion of the genomes extracted from *N. albigula* that had at least one oxalate-related gene. (**B**) Relative distribution of oxalate-handling genes by gene function. (**C**) Proportion of genomes that have a complete pathway for oxalate degradation, specifically. oxdd = oxalate oxidoreductase; acec = acetyl-CoA:oxalate CoA-transferase; oxc = Oxalyl CoA decarboxylase; succ = succinyl-CoA:coenzyme A transferase; frc = Formyl CoA transferase; (**D**) Number of oxalate-handling genes by genome.

## Oxalate metabolism is driven both by substrate availability and microbiota composition

Adaptability and homeostatic feedback within complex systems is driven by the convergence of system components and resource availability (*Cohen et al., 2022*). To examine the confluence of resource availability and oxalate metabolism, a custom medium based on previously published gut microbiota media (*Goodman et al., 2011*) was modified by adding substrates associated with metabolic pathways enriched by oxalate (*Figure 3C*, *Supplementary file 6*). The oxalate-degrading species *Enterococcus gallinarum,* previously isolated from NALB (*Miller et al., 2014*) and the whole NALB community were assessed. Chosen substrates impacted oxalate metabolism and the impact of oxalate on growth, particularly at the community level (*Figure 5A and B*). A minimal media with the same substrates added as sole carbon and energy sources (*Supplementary file 7*) allowed for quantification of the proportion of the NALB microbiota that could use each substrate as sole carbon and energy sources. Culturomic data recapitulates molecular data to show a considerable amount of redundancy surrounding oxalate metabolism (*Figure 5C*). Isolates generated from this assay were used for subsequent study (metabolic cohort; *Figure 5D*). Additionally, a second cohort was defined and commercially purchased based both on known metabolic functions and the proportion of studies that saw an increase in their taxonomic population with oxalate consumption (*Figure 5D*; taxonomic cohort). Where possible, isolates from human sources were obtained. Cohorts, defined in the STAR methods, were used to delineate hypotheses that either carbon and energy substrates are sufficient to explain known effects of the oxalate-degrading microbial network or that additional aspects of taxa commonly stimulated by dietary oxalate are required to explain past results (taxa defined through previous meta-analysis of studies) (*Batagello et al., 2018*). Oxalate metabolism with the metabolic and taxonomic cohorts was assessed in vitro in minimal media with oxalate as a sole carbon and energy source (*Figure 5E and F*). There were considerable differences in oxalate metabolism in both cohorts, dependent on the microbes present. However, significant oxalate metabolism occurred even in the absence of *O. formigenes,* indicative of metabolic redundancy. Collectively, data show that both resource availability and community composition impacts oxalate metabolism, which helps to define the adaptive nature of the NALB gut microbiota. Additionally, results further bolster evidence for redundancy surrounding oxalate metabolism.

## Severity of oxalate-induced microbe-host effects is dependent on microbial oxalate metabolism, independent of taxonomy

To delineate hypotheses of metabolic redundancy or cooperation for mitigating the negative effects of oxalate on the gut microbiota and host, two independent diet trials were conducted with analogous microbial communities derived from the metabolic and taxonomic cohorts. Following antibiotic suppression of the gut microbiota, SWM were given microbial transplants from either the metabolic or taxonomic cohorts in a longitudinal, crossover experimental design with either a 0 or 3% oxalate diet (Envigo; *Figure 6A*, *Figure 6—figure supplement 3A*; *Supplementary file 1*). The 0% oxalate diet was designed to test the stability of oxalate-degraders since *O. formigenes* is often lost when oxalate is removed from the diet (*Batagello et al., 2018*). Transplanted microbial communities included the same as those for in vitro studies (*Figure 5D–F*). Animal metrics and microbial were tracked over the course of the trial as was urinary/fecal oxalate, urinary formate, inflammatory cytokines, and creatinine. Renal calcium oxalate (CaOx) deposition, cardiac fibrosis, and colitis was assessed through histopathology. CaOx deposition and cardiac fibrosis was quantified through a semi-automated process, based on stain color that differentiates calcium deposits (Von Kossa) or collagen (Mason's Trichrome). Colitis severity was assessed by two independent reviewers in hematoxylin and eosin stained tissues through a standardized, multi-factorial assessment (*Koelink et al., 2018*).

Using microorganisms from the taxonomic cohort, while the change in urinary and fecal oxalate levels were greatest in mice given *O. formigenes*, the change in oxalate levels were significantly greater than the no bacteria controls (Group 1) even in the absence of *O. formigenes,* consistent with in vitro results (*Figure 6B and C*). Interestingly, the change in urinary formate levels was not different in any microbial group for the taxonomic cohort (*Figure 6D*). While IL1β was below detection levels, IL6 exhibited levels consistent with oxalate induction, which decreased over the course of the trial (*Figure 6B*, *Figure 6—figure supplement 1A*). Differences in IL18 were only seen as an increase over time in the No_ox (Group 5) (*Figure 6—figure supplement 1B*). The only changes in urinary creatinine

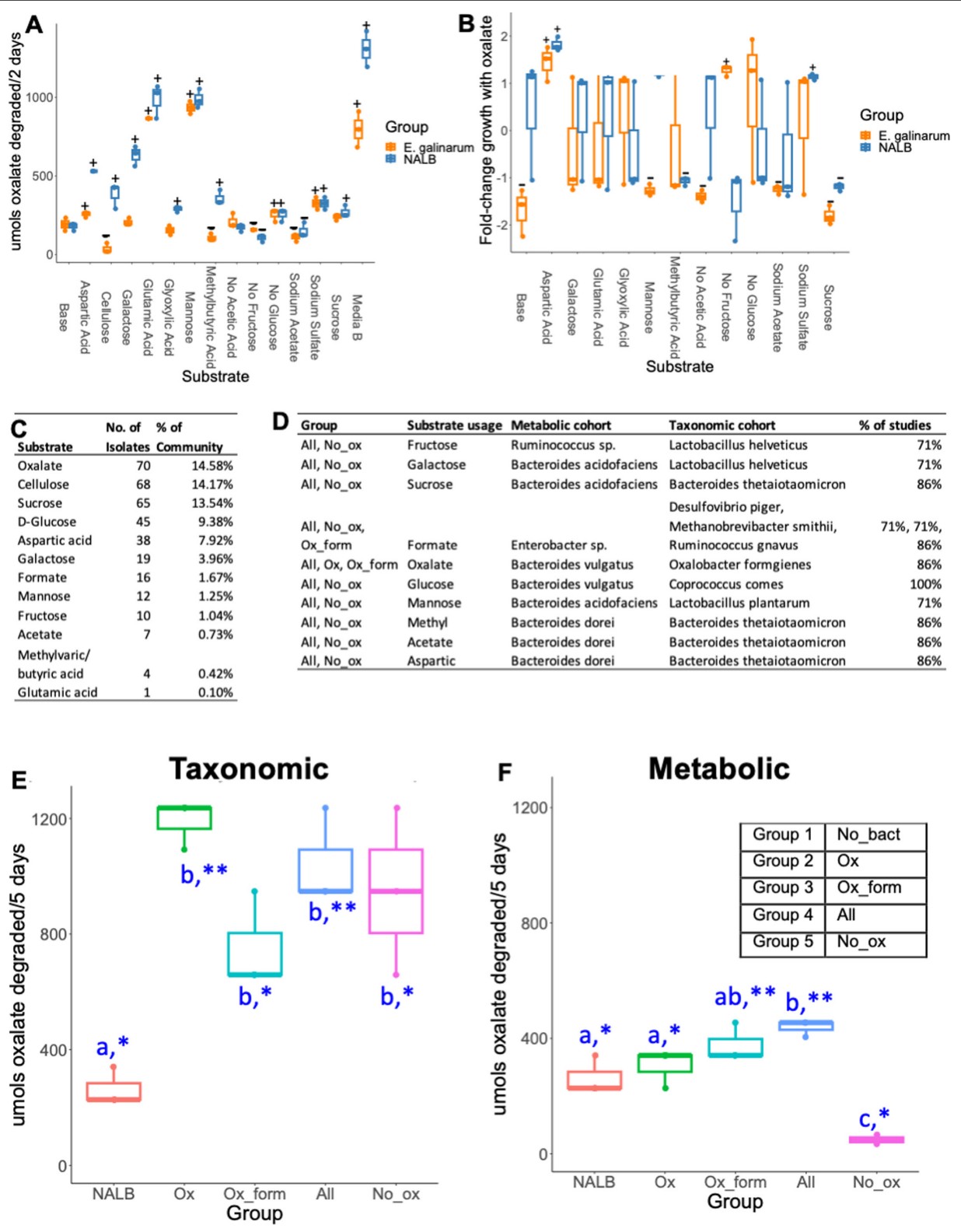

**Figure 5.** Microbial community composition and available substrates impact oxalate metabolism and the impact of oxalate on growth. (**A**) Substrates associated with metabolic pathways enriched by exposure to dietary oxalate in vivo differentially impact oxalate metabolism. p<0.001 in one-way and two-way ANOVA against bacterial group and substrate; +/- reflects p<0.05, Holm's-corrected, pairwise t-test compared to base media for an increase (+) or decrease (-) in oxalate degradation. (**B**) Substrates differentially impact the influence of oxalate on microbial growth; p<0.001 in two-way ANOVA

*Figure 5 continued on next page*

*Figure 5 continued*

against substrate and bacterial group, and one-way analysis against substrate; +/- reflects p<0.05, Holm's-corrected, one-sample t-test compared 0 (no impact of oxalate) for an increase (+) or decrease (-) in growth due to oxalate exposure. The impact of oxalate on growth was not calculated for cellulose or Media B. (**C**) Culture-based means to quantify proportion of the NALB community that can use substrates identified through shotgun metagenomics as sole carbon and energy sources. (**D**) Defined microbial communities to assess oxalate metabolism in vitro and in vivo. Listed are the microbial consortia, which substrate the microbes utilize that corresponds to the shotgun metagenomic data, taxonomic classification of microorganisms used in the two cohorts, and the proportion of studies in which microorganisms in the taxonomic cohort were stimulated by oxalate exposure. (**E,F**) Oxalate metabolism in minimal media with 20 mM oxalate from the microbial communities listed in 5D, in comparison to the *N. albigula* (NALB community). p<0.001, ANOVA comparing microbial groups. *p<0.05, **p<0.01; ***p<0.001; Holm's corrected, one sample t-tests against 0 (no oxalate metabolism). Blue letters reflect statistical groups between microbial groups for oxalate metabolism. N=3 for all experiments.

seen was an increase and decrease in the NALB and All (Group 4) (***Figure 6—figure supplement 1C***), respectively, which indicates that while the NALB bacteria may induce some inflammation, the minimal community present in the All (Group 4) group may limit these effects and actually improve kidney health under conditions of oxalate exposure. While some differences were seen in water or food metrics, these can largely be explained by batch effects of the two trials conducted (***Figure 6—figure supplement 1D–G***). We did see a greater increase in body mass in animals receiving the No_ox (Group 5) microorganisms than when *O. formigenes* was administered alone (Group 2), which can not be explained by batch effects. Renal CaOx deposition, cardiac fibrosis, and colitis severity all closely tracked oxalate levels (***Figure 6E-J***, ***Figure 6—figure supplement 1H-I***) and did not depend on *O. formigenes* if the other probiotic microbes were present. Collectively, data suggest that oxalate homeostasis and oxalate-induced pathologies are mitigated by the presence of diverse oxalate-degrading bacteria and that adding *O. formigenes* on an already effective oxalate-degrading microbiota will not improve oxalate homeostasis.

Based on high-throughput 16 S rRNA sequencing, the microbiota composition of colon contents from this diet trial was significantly different from excreted feces (***Figure 6***) The microbiome composition in stool from mice was not different based on probiotic administration (***Figure 6—figure supplement 2B***), but within-group differences for the All (Group 4) group was significantly lower than other groups (***Figure 6—figure supplement 2C***), indicative of a protective effect against oxalate exposure. This conclusion is corroborated by the change in alpha diversity in which the microbiota of animals given the All (Group 4) group exhibited the greatest post-antibiotic recovery and is the only group that saw recovery beyond a marginally significant increase in diversity, post-antibiotics (***Figure 6—figure supplement 2D***). In general, there was not much loss of the inoculated bacteria throughout the study, in any group (***Figure 6—figure supplement 2E***).

Overall, metabolic cohort transplants were less effective than the taxonomic cohort in terms of inducing oxalate metabolism, particularly in the group without bacteria (Group 1) isolated against oxalate (***Figure 6B, C***, ***Figure 6—figure supplement 3B, C***). However, there was a much greater change in format metabolism, dependent on the microbes present (***Figure 6—figure supplement 3D***). Despite the more moderate change in oxalate levels, there were similar oxalate-associated effects on renal calcium oxalate deposition (***Figure 6—figure supplement 1F, G***), which was not dependent on formate levels (***Figure 6—figure supplement 3***). With the metabolic cohorts, the presence of known oxalate degraders appeared to be more important for oxalate homeostasis than the taxonomic cohort, since the absence of known oxalate-degraders led to oxalate levels similar to the No_bact (Group 1) controls (***Figure 6B and C***; ***Figure 6—figure supplement 1B, C***). Collectively, data show that while the taxa chosen for the taxonomic cohort enabled efficient oxalate even in the absence of *O. formigenes*, the bacteria present in the metabolic cohort had a greater influence on formate levels. In both diet trials, urinary oxalate, but not formate correlated with CaOx deposition (***Figure 6—figure supplement 3G and H*** – metabolic, data not shown - taxonomic). The effects of oxalate and transplant group differed between the two cohort studies with mice in the metabolic cohort exhibiting higher levels of IL-6 of microbial transplant groups compared to the negative controls. However, similar to the taxonomic cohort, there were no differences in IL18. Urinary creatinine increased significantly over time for all groups receiving a microbial transplant, in contrast with the taxonomic cohort (***Figure 6—figure supplement 1A–C***, ***Figure 6—figure supplement 4A–C***). For the metabolic cohort, there are similar group-based differences observed in water-based metrics. However, animals in the taxonomic

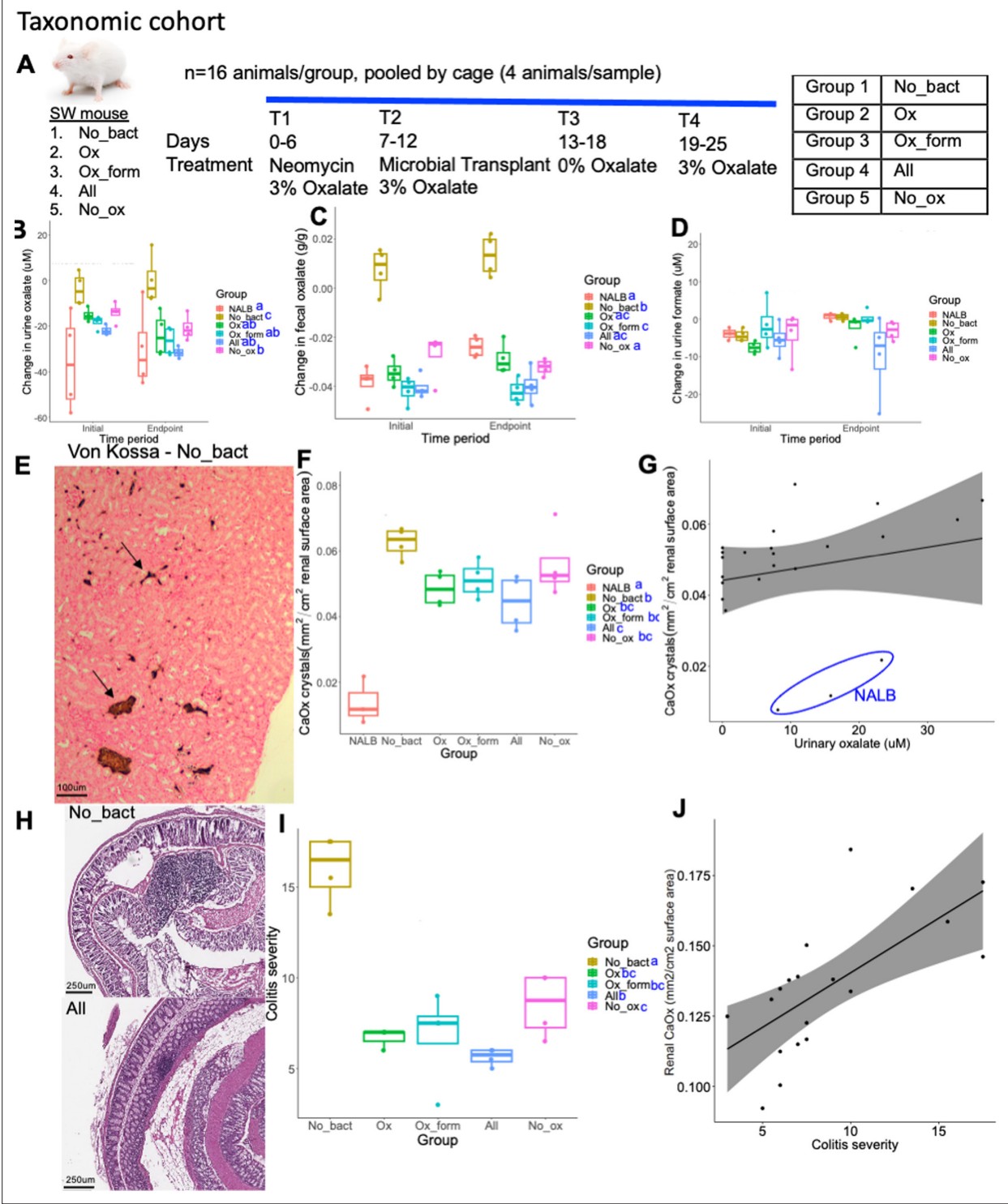

**Figure 6.** Microbial community composition (taxonomic cohort) impacted the effect of exogenous oxalate on host health. (**A**) Swiss Webster mice were given neomycin, followed by inoculation of microbial consortia that included either no bacteria or the taxonomic cohort listed in *Figure 5C*. (**B,C**) The effect of microbial transplants on urinary (**B**) or fecal (**C**) oxalate levels over the course of the diet trial, compared to baseline. ANOVA p<0.001 for microbial group, but was not significant by time period or two-way analyses for both B & C. (**D**) The effect of microbial transplants on urinary formate levels over the course of the diet trial, compared to baseline. ANOVA was not significant in one-way and two-way analyses. (**E**) Renal calcium oxalate deposition. Arrows show stained calcium deposits, which were quantified through an automated algorithm in QuPath. (**F**) Quantification of renal calcium oxalate deposition by group. p<0.001, ANOVA. Blue letters reflect statistical groups between microbial groups for renal calcification by Holm's corrected paired t-tests. (**G**) Pearson correlation between urinary oxalate and renal calcium oxalate deposition. *R*=0.22, p=0.32 with *N. albigula* (NALB)

*Figure 6 continued on next page*

*Figure 6 continued*

samples included (blue circle); *R*=0.7, p=0.001 excluding the NALB group. (**H**) Representative colon tissues from the No_bact (Group 1) and All (Group 4) groups, exhibiting high and low colitis severity scores, respectively. Tissues were stained with hematoxylin and eosin and scored based on standardized, multifactorial metrics. (**I**) Quantification of colitis severity by group. p<0.001, ANOVA. Blue letters reflect statistical groups between microbial groups for renal calcification by Holm's corrected paired t-tests. (**J**) Pearson correlation between colitis severity and renal calcium oxalate deposition. *R*=0.7, p=0.002. Colitis severity was not quantified for the NALB group.

The online version of this article includes the following figure supplement(s) for figure 6:

**Figure supplement 1.** Microbial transplant composition impact on urinary inflammation, renal health, and overall mouse health for the taxonomic cohort.

**Figure supplement 2.** Effect of microbial transplants on microbial community composition for the taxonomic cohort.

**Figure supplement 3.** Microbial community composition (metabolic cohort) impacted the effect of exogenous oxalate on kidney health.

**Figure supplement 4.** Microbial transplant composition impact on urinary inflammation, renal health, and overall mouse health for the metabolic cohort.

**Figure supplement 5.** Effect of microbial transplants on microbial community composition for the metabolic cohort.

**Figure supplement 6.** Effect of oxalate and formate on oxalate quantification using an enzymatic, ELISA-based assay.

cohort exhibited greater positive changes in food intake and body mass than the metabolic cohort (*Figure 6—figure supplement 4D–G*).

In the metabolic cohort, we did not observe the same trends in the microbiota composition, antibiotic recovery, or persistence as we did in the taxonomic trial indicative of less effective microbial communities at protecting the community as a whole (*Figure 6—figure supplement 5A–D*), though probiotic bacteria were generally persistent (*Figure 6—figure supplement 5E*). Data were derived from high throughput sequencing of the 16 S rRNA gene.

Overall, data from the taxonomic and metabolic cohorts indicate that oxalate metabolism results from a defined community of microorganisms that includes redundancy in oxalate metabolism as the primary driver. Importantly, the efficacy of *O. formigenes* was apparent when administered alone, but the effect was diminished when co-inoculated with other oxalate-degrading microbes.

## Discussion

While research in the last two decades have made clear that the gut microbiota is intimately tied to all facets of host health, translating those insights into actionable biotherapies has been difficult due to the inherent complexity and heterogeneity present in the microbe-host system. The study of complex systems holds enormous potential to constrain hypotheses and offer insight into effective bacteriotherapy development (*Lambiotte et al., 2019*).

However, the field is limited by inconsistent terminologies, concepts, and tractable experimental frameworks (*Torres et al., 2021*.) The objective of the current study was to examine oxalate-microbe-host interactions within a complex systems framework. Collectively, the results of the study offer quantifiable and generalizable metrics of diet-microbe-host systems that can be used to guide more effective patient and probiotic selection criteria for clinical trials involving bacteriotherapies that target the gut microbiota. Specifically, with complex system modeling, we found through multiple in vivo and in vitro experimental systems, that exogenous oxalate exhibits systemic effects on multiple organs, including the gut microbiota, and oxalate homeostasis is specifically mitigated by diverse oxalate-degrading bacteria rather than a single bacterial species or factors such as host species, or microbe-microbe cooperation (*Figure 7*).

Using our complex modeling approach, our findings establish that oxalate's effects on both the host and the gut microbiota are primarily mediated by the metabolic redundancy of oxalate metabolism in the gut microbiota, rather than a single species or other factors. Specifically, experiments in *Figure 1* demonstrated that host responses to oxalate were dependent on the composition of the gut microbiota. Expanding on this, experiments in *Figure 2* revealed that oxalate's impact on microbial community structure was similarly reliant on the whole microbiota, and direct multi-omic data integration confirmed that these microbial changes subsequently influenced host responses to oxalate. Once the gut microbiota composition was deemed to be critical, experiments in *Figure 3* examined the specific oxalate impacts on the gut microbiota, using *N. albigula* and its native microbiota as the

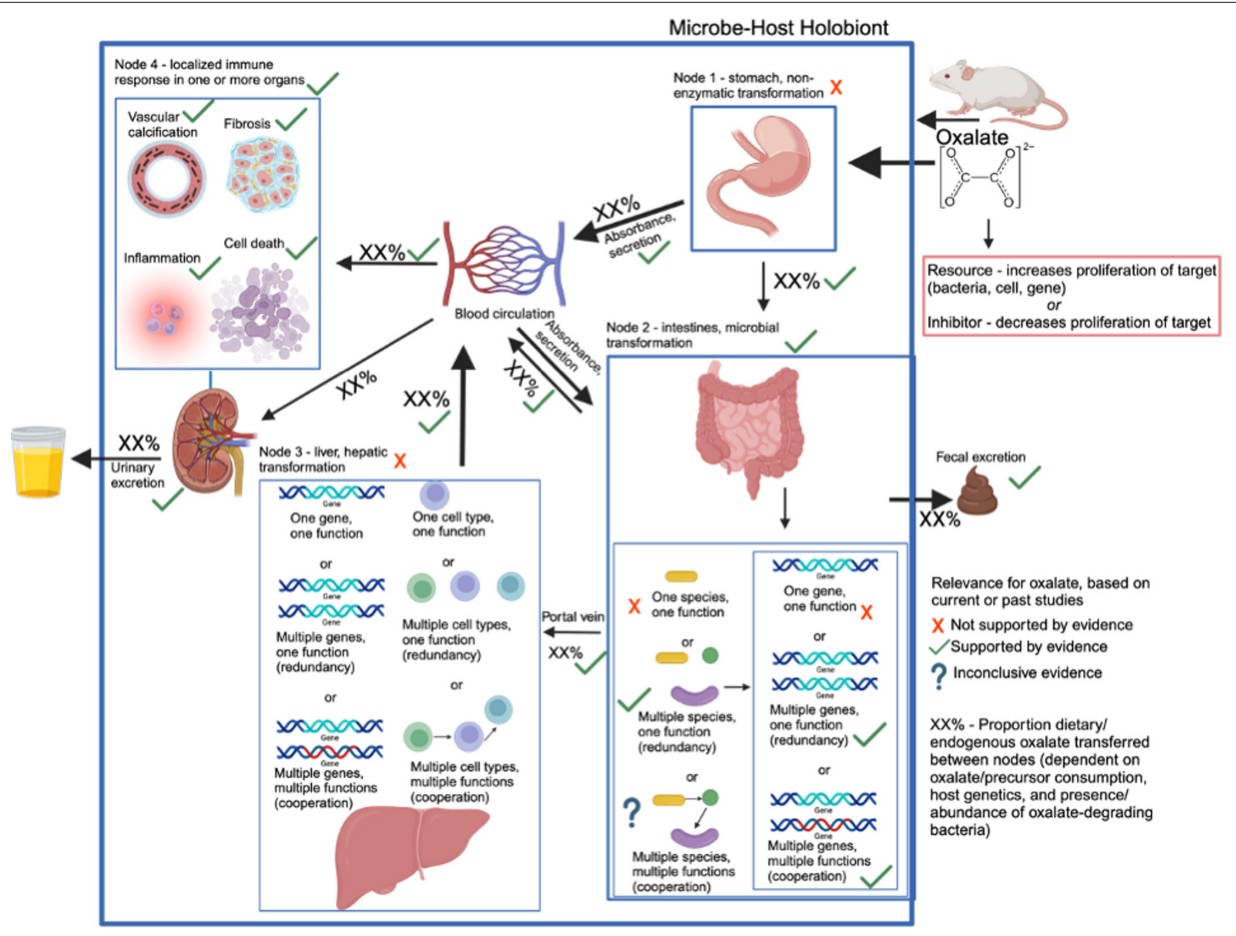

**Figure 7.** Model of oxalate-host-microbiome interactions, based on experiments here. Model starts in the top left and proceeds along arrows through fecal or urinary excretion. Included in the model are the processing nodes, the potential hypotheses on cooperative vs. redundant functions, and which hypotheses our data favor or refute. Red x's are hypotheses refuted by our data, green arrows are those confirmed by our data, and question marks reflect hypotheses where evidence is inconclusive.

model for high oxalate tolerance. This analysis identified multiple genes associated with oxalate degradation, as well as acetogenic, methanogenic, and sulfate-reducing pathways. To assess the functional redundancy of these pathways, *Figure 4* and *Supplementary files 2-5* quantified the extent to which multiple microbial species contribute to these pathways. The high degree of redundancy observed suggested that oxalate degradation is maintained through diverse oxalate-degrading bacteria rather than a single keystone species. To explore the environmental factors influencing oxalate metabolism, we employed a culturomic approach to characterize microbial responses under controlled conditions and to evaluate oxalate degradation within our defined, hypothesized microbial communities (based on *Figure 3* data and past oxalate diet trials). Finally, *Figure 6* validated our metagenomic, metabolomic, and culturomic findings through targeted microbial transplants in mice, specifically demonstrating that the diversity of oxalate-degrading microbes were transferable and influenced host oxalate homeostasis more than a single species. The overall complex modeling approach was designed to systematically identify the most critical factors in the oxalate-microbiota-host relationship. Each experiment was informed by prior results, allowing for a stepwise refinement of hypotheses and experimental design. This integrative framework underscores the importance of metabolic redundancy in modulating oxalate metabolism and its downstream effects on host physiology.

Microbial oxalate metabolism, which has been researched for several decades, pre-dating the current microbiome era, is an ideal focal point to understand pertinent variables that impact probiotic success. Here, we found that heterogeneity in microbiome composition, independent of host genetics, significantly impacted multiple diet-microbe-host interfaces that included hepatic activity

(*Figures 1 and 2*), gut microbiota metabolism (*Figures 2 and 3*), renal mineralization (*Figure 6E and F*; *Figure 6—figure supplement 3E and F*), intestinal inflammation (*Figure 6H,I*), and cardiac fibrosis (*Figure 6—figure supplement 1H1*). Importantly, we found that oxalate degradation was highly redundant among diverse species in the gut of *N. albigula*, which consumes a high oxalate diet in the wild (*Figure 4*). Consistent with past studies, administering *O. formigenes* led to a significant reduction in both stool and urine oxalate (*Figure 6B and C*). However, when co-administered with other oxalate-degrading microorganisms, there was no additive effect. Oxalate-based results were similar, but to a lower degree, in the metabolic cohort (*Figure 6—figure supplement 3B and C*). In taxonomic cohort studies, we did not see significant differences in the change in urine formate, based on the microbial transplant group (*Figure 6D*), in contrast to metabolic cohort studies (*Figure 6—figure supplement 3D*).

Collectively, data offer strong support for the hypothesis that metabolic redundancy among diverse taxa, is the primary driver of oxalate homeostasis, rather than metabolic cooperation in which the by-products of oxalate degradation are used in downstream pathways such as acetogenesis, methanogenesis, and sulfate reduction. However, data on the metabolic cooperation hypothesis were inconclusive and there are multiple known microbe and host sources of formate, which may confound results in these studies (*Asanuma et al., 1999*; *Hughes et al., 2017*; *Pietzke et al., 2020*; *Ternes et al., 2022*).

Modeling the microbe-host system through an evolutionary perspective, it has previously been suggested that while host control over the microbiome is the primary driver of the form and function of the gut microbiota, given the enormous biomass and diversity of the gut microbiota, microbe-microbe competition is also required to constrain the microbiota in mammalian microbe-host systems (*Foster et al., 2017*). In the current (*Figures 2B and 5B*) and previous studies (*Allison et al., 1985*; *Miller and Dearing, 2013*; *Turroni et al., 2010*; *Turroni et al., 2007*), we and others have found that oxalate can differentially exhibit positive or negative effects on microbial growth and metabolism dependent on the species and environment present. These data provide two alternative ecological pressures to degrade oxalate. The first is to use oxalate as a carbon and energy source for growth, as is the case with *O. formigenes* (*Allison et al., 1985*). The second is to degrade oxalate to remove it as a toxin, as is the case with some *Lactobacilli* and *Bifidobacteria* (*Turroni et al., 2010*; *Turroni et al., 2007*). The data showing oxalate degradation as a metabolically redundant function among multiple diverse microorganisms that maintain oxalate homeostasis (*Figures 4 and 6*, *Figure 6—figure supplement 3*), suggest that the amount of oxalate consumed or produced by the liver is too great for a single, slow-growing oxalate-degrading specialist can handle alone, in contrast to prevailing hypotheses (*Daniel et al., 2021*), and in support of the ecosystem on a leash hypothesis. Importantly, the near-universal presence of formate metabolism genes suggests that formate may be an even greater source of ecological pressure (*Figure 3—figure supplement 2*, *Figure 3—figure supplement 5*). In fact, multiple studies have shown that formate is toxic both to many bacterial species and to mammalian hosts, which supports this hypothesis (*Hughes et al., 2017*; *Pietzke et al., 2020*; *Ternes et al., 2022*; *Kirkpatrick et al., 2001*; *Voskuhl et al., 2022*). Collectively, data from the current and previous studies on the effect of oxalate exposure on the gut microbiota (*Miller et al., 2017a*; *Miller et al., 2016b*) support the hypothesis that the gut microbiota serves as an adaptive organ (*Scott et al., 2013*; *Doré and Blottière, 2015*; *Britton and Young, 2014*) in which specific, metabolically redundant microbes respond to and eliminate dietary components, for the benefit of themselves, but which can residually protect or harm host health depending on the dietary molecules and gut microbiota composition (*Tang and Hazen, 2024*; *Scott et al., 2013*; *Doré and Blottière, 2015*; *Britton and Young, 2014*).

Oxalate degradation, as a focal point in diet-microbe-host interactions, is a special case in which the effects of oxalate on host health and the mechanisms of microbial oxalate degradation have largely been worked out for decades (*Allison et al., 1985*; *James and Butcher, 1972*; *Anantharam et al., 1989*; *Baetz and Allison, 1989*; *Baetz and Allison, 1990*). Furthermore, oxalate degradation exhibits moderate complexity, both in that it is isolated to the gut microbiota and is performed through a handful of simple metabolic pathways (*Cochat and Rumsby, 2013*; *Hodgkinson, 1977*; *Allison et al., 1985*; *Miller and Dearing, 2013*; *Liu et al., 2021a*). Despite this knowledge, clinical trials designed to reduce systemic oxalate using either *O. formigenes* or other oxalate-degrading bacteria have exhibited a wide variety of patient and trial responses (*Batagello et al., 2018*). However, knowledge about

other diet-microbe-host links are largely in their infancy. Many of the other known features exhibit much greater complexity than oxalate degradation. For instance, while the production of short-chain fatty acids are widely viewed as beneficial for both the gut microbiota and host health (*Deleu et al., 2021*; *Kelly et al., 2015*; *Portincasa et al., 2022*; *Shimizu et al., 2019*; *Tian et al., 2018*; *Vieira et al., 2017*), many short-chain fatty acids are known and are produced by a wide variety of species (*Fusco et al., 2023*; *Mann et al., 2024*). Other important diet-microbe-host links, such as the production of secondary bile acids or TMAO (*Tang and Hazen, 2024*; *Tang et al., 2015*; *Chiang, 2009*; *Collins et al., 2023*; *Gérard, 2013*), also involve host hepatic activity, increasing the complexity further. Given the complexity of diet-microbe-host interactions and the fact that understanding the mechanisms through which the gut microbiota modifies the diet to influence host health is demonstrably insufficient for the development of effective bacteriotherapies (*Barkhidarian et al., 2021*; *Rogers and Mousa, 2012*; *Ceccherini et al., 2022*; *Batagello et al., 2018*), we propose a three-phase, preclinical experimental workflow for the development of targeted bacteriotherapies. In the first phase of research, case:control metagenome-wide studies and research into other factors that modify the microbiome, such as prior antibiotic exposure, can identify disease phenotypes that are influenced by the microbiome, along with the microbial taxa and genes responsible for this association. In the second phase of research, hypothesized taxa and/or metabolic functions can be explored through in vitro and germ-free animal studies, using appropriate models to determine the mechanistic links that drive disease phenotypes through diet-microbe-host interactions. Finally, in phase three, these mechanistic links can be applied to a complex system theoretical framework, as done in the current study, to identify those variables most pertinent to successfully influence specific phenotypes through targeted changes to the gut microbiota. Such a phased research structure will provide for much more effective probiotic and patient selection criteria prior to clinical trials.

There were some important limitations to the current study. First, while we did conduct one study with *N. albigula,* most animal studies here sought to eliminate host genetics from the equation, as a means of simplification. Host genetics are another layer of complexity that were not examined here and may have an impact on oxalate homeostasis beyond the gut microbiota. Second, in animal studies with the refined communities of microorganisms, the taxonomic and metabolic cohorts studies were conducted separately and some batch effects are apparent, such as with the change in creatinine values or urine output. Finally, it is apparent from our data and the literature that there are multiple host and microbial sources of formate. As such, data pertaining to formate utilization by our transplant communities are inconclusive.

In conclusion, using a complex systems theoretical framework, we examined the oxalate-microbe-host interactions of multiple oxalate-microbe-host interfaces and found multiple microbiome-dependent effects of oxalate. The negative effects of oxalate were mitigated by metabolically redundant oxalate-degrading bacteria, more so than by metabolic cooperation or a single oxalate-degrading species. Critically, we found that while *O. formigenes* can lower urine oxalate when placed on a background of a poor oxalate-degrading community, as current theory predicts, this effect is lost when co-administered with other oxalate-degrading bacteria. Collectively, data help to resolve why the gut microbiota of the white-throated woodrat, *N. albigula,* effectively responds to and degrades even very high dietary oxalate levels (*Miller et al., 2017a*; *Miller et al., 2017b*; *Miller et al., 2016a*) and offer a clear pathway for more effective patient and probiotic selection for future clinical trials to reduce urine oxalate. More importantly, the conceptual and experimental framework developed in this study, based on complex systems theory, paves the way for a phased approach to microbiome research in which clinical microbiome insights (Phase I) drive mechanistic insights between the gut microbiota and host physiology (Phase II), that can be applied to a complex system model to constrain hypotheses and identify the most pertinent variables that drive microbe-host interactions that influence host physiology and health (Phase III). Such an approach will allow for much more efficacious probiotics and patient selection criteria in clinical intervention trials.

## Methods

### Animal studies

#### Tracking animal health

Throughout all diet trials, animals were monitored for evidence of trauma, dehydration, pain, or other forms of suffering. Additionally, water and food intake, urine output, and body mass was monitored daily (i.e. *Supplementary files 6 and 9*). If animals lost >10% of their baseline body mass, they were removed from the diet trial and placed back onto standard mouse chow.

### Effect of oxalate on the native gut microbiota of *N. albigula*

To determine the impact of dietary oxalate on *N. albigula* with their native microbiota, 14 adult *N. albigula* woodrats were collected from Castle Valley, UT (38.63′N, 109.41′W) in September 2014, using Sherman live traps. Animals were transported to the University of Utah and housed in individual cages (48 by 27 by 20 cm) with a 12 hr/12 hr light/dark cycle, at 28 °C and 20% humidity. Animals were initially maintained on a 0.2% oxalate, high-fiber rabbit chow (Harlan Teklad formula 2031; Envigo) for 7 mo prior to experimentation. This diet reduces the overall detectable diversity of the microbiota of these animals but maintains the members of the native microbiota overall (*Kohl and Dearing, 2014*). All methods were approved by the IACUC under protocol no. 12–12010.

Subsequently, *N. albigula* woodrats were placed in a diet trial in which the oxalate concentration of the food was gradually increased over time along a gradient from 0.2% (for days 1–5; herein referred to as 0%), 3% (days 6–10), and 6% (days 11–15; *Figure 3A*). The oxalate concentration of the diet was adjusted by adding the appropriate amount of sodium oxalate (Fisher Scientific, Pittsburgh, PA) into the powdered rabbit chow on a dry weight basis. The schedule for each concentration of dietary oxalate was chosen to ensure that the gut microbiota had time to respond to the specific diet (*Belenguer et al., 2013*; *Miller et al., 2016b*). During diet trials, oxalate did not have a negative impact on *N. albigula* health and neither fecal nor urinary oxalate levels increased significantly (*Miller et al., 2017a*).

During the diet trial, animals were placed in metabolic cages to separate urine and feces into sterile 50 ml conical tubes. A subsample of feces was collected from each animal every 4–5 d on the 0% and 6% oxalate diets for shotgun metagenomic sequencing. Samples were frozen at −80 °C until DNA extraction.

### Stool collection for bacterial isolations and fecal transplant studies

Feces for fecal transplants were collected from two sources. For the Swiss-Webster feces, 20 individual animals (Taconic Farms, female, 6-wk-old) were placed on a 0% oxalate diet in cages with a custom-designed insert to separate urine and feces at the Cleveland Clinic (IACUC #2016–1653) (*Miller et al., 2019b*). For the *N. albigula* feces, 14 individual animals (mixed sex and age) were placed on a 3% oxalate diet in a metabolic chamber at the University of Utah (IACUC #12–12010) to acclimate the microbiome to oxalate metabolism (*Miller et al., 2019b*). After 3 d of acclimation on the respective diets, feces were collected from animals within 2 hr of defecation, submerged in sterile 15% glycerol, and flushed with $CO_2$ prior to freezing at −80 °C until use in animal studies approximately 12 mo after receipt (IACUC #2016–1653) or for bacterial isolations.

### Microbiota-dependent effects of oxalate on liver and gut microbiota

All animal studies were approved through the Cleveland Clinic's IACUC (IACUC #'s 2016–1653, 2020–2312). To examine the host-independent effects of oxalate on gut microbiota with either a low or high efficiency at degrading oxalate, two distinct host-microbe model systems were developed. For both models, equal numbers of male and female Swiss-Webster mice were used as the host. For the low oxalate-degrading gut microbiota, Swiss-Webster mice were given fecal transplants using feces from other Swiss-Webster mice (allografts), as discussed below. A 1.5% oxalate diet is sufficient to induce hyperoxaluria in Swiss-Webster mice, defined as a 50% increase in urinary oxalate excretion (*Miller et al., 2017b*; *Miller et al., 2016a*; *Hatch et al., 2011*). For the high oxalate-degrading gut microbiota, Swiss-Webster mice were given fecal transplants from *N. albigula* (xenografts). We have previously shown that fecal transplants from *N. albigula* are sufficient to induce significant and persistent oxalate metabolism (*Miller et al., 2017b*; *Miller et al., 2016a*). During and after fecal transplants,

animals were grouped into four to a cages, with four cages assigned to each treatment group, for a total of 16 animals per experimental group. To minimize individual variability and eliminate any cage effect in subsequent metrics, samples and data collected from all animals in a cage were pooled together into an individual sample. To develop the two different mouse models, the conversion of the gut microbiota from the native microbiota to the grafted microbiota (allografts or xenografts) was completed in two stages. First, the native microbiota was depleted with a 5-d course of neomycin (0.5 g/L water, combined with 2 g/L sucralose), while on a 0% oxalate diet. Mice were given ad libitum access to water during this time and throughout the experiment. Neomycin is a broad spectrum antibiotic that is poorly absorbed across the gut, and effectively reduces gut microbiota density by up to 90% (*Vijay-Kumar et al., 2010*). After antibiotic depletion, mice were given the respective fecal transplants by first thawing feces, then aseptically mixing 32 g feces per kg body weight directly into powdered mouse chow. Fecal transplants were performed daily for 6 d (*Figure 1A*), as previously described (*Miller et al., 2019b*), during which animals were switched to either a 0 or 1.5% oxalate diet. All animals were maintained on their respective diets for 2 wk following fecal transplants (*Supplementary file 1*). A total of four treatment groups were defined based on allografts/xenografts and oxalate diet.

Following diet trials, animals were sacrificed through $CO_2$ asphyxiation and cervical dislocation. Upon necropsies, liver tissue from each animal was dissected out and placed into RNA later on ice, prior to freezing at –80 °C, within 2 hr of necropsies. Additionally, colon feces were collected and placed into screw-cap tubes with o-rings, on ice, prior to freezing at –80 °C.

## Microbial cohort-dependent effects on mitigating the systemic effects of oxalate on host and microbiome

To determine the effect of specific microbial cohorts at mitigating the systemic host-microbiome effects of oxalate, different combinations of two microbial cohorts were established. First, based on the metabolic pathways enriched by oxalate in the *N. albigula* gut microbiota (*Figures 3D and 5C*, *Supplementary file 5*), assessed through comparative shotgun metagenomics (described below), we hypothesized that microorganisms that engage in those metabolic pathways help to mitigate the systemic effects of oxalate. To test this hypothesis, we isolated bacteria from *N. albigula* stool (described below) and grouped microorganisms into four cohorts (*Figure 5D*). These included oxalate-degrading bacteria alone (Ox; Group 2), microorganisms that can utilize formate or oxalate (Ox_form; Group 3), oxalate and formate users, in addition to the metabolism of selected sugars (All; Group 4), or the complete cohort except for oxalate degraders (No_ox; Group 5). Microorganisms isolated from *N. albigula* stool and used in microbial transplants were termed the 'metabolic cohort.' In addition to the metabolic cohort, a 'taxonomic cohort' was defined by cross-referenced metabolic pathways enriched by oxalate with microbial taxa previously reported to be stimulated by oxalate (*Figure 5D*; *Batagello et al., 2018*). Microorganisms from this cohort were purchased commercially from the ATCC or DSMZ (*Supplementary file 8*). In addition to these cohorts, we also examined a negative control in which animals were inoculated with sterile media (No_bact (Group 1); Group 1), and a positive control in which animals were inoculated with the whole gut microbiota of *N. albigula*, extracted from a stool by first vortexing in a 1:1 mixture of stool and sterile PBS, followed by centrifuging at 8000 RPM for 2 min. The supernatant was inoculated into gut microbiota media with 20 mM sodium oxalate (*Supplementary file 6*) overnight prior to microbial transplant.

To prepare each isolate for transplants, an overnight culture of each species was grown and mixed in equal proportions by volume for each cohort. The community was then centrifuged and the supernatant was decanted. After antibiotic depletion of the native gut microbiota in Swiss-Webster mice, 38 mg of pelleted bacteria of each preparation was added directly to the food and administered over the course of 6 d (*Figure 6*, *Supplementary file 8*), as done previously (*Miller et al., 2017b*; *Hatch et al., 2011*). Preparations corresponded to approximately $5 \times 10^8$ microorganisms for each isolate in the appropriate transplant groups. Quantification of bacterial numbers was performed through absorbance of the microbial preparations on a spectrophotometer at 600 nm, calibrated to direct microscopic counts.

As above, animals were grouped four into a cage, with samples from each cage pooled together prior to collection and analyses. Similarly, animals were maintained on custom inserts that mimic metabolic cages for the effective separation of urine and stool prior to collection. During the microbial

transplant animal studies, stool and urine was collected daily and inserts were disinfected. For oxalate and formate quantification, samples were collected at the end of the antibiotic depletion, microbial transplants, a 0% oxalate washout period, and after return to a 3% oxalate diet (*Figure 6B–D*, S8B-D). Prior to biochemical assays, urine was frozen (−20 °C) and feces were dried at 45 °C overnight. Prior to freezing, aliquots of urine were equally divided into six fractions for each of the biochemical assays performed (discussed below). Fractions for oxalate quantification were collected into 4 N HCl prior to freezing to prevent the non-enzymatic conversion of ascorbic acid into oxalate (*Miller et al., 2016b*). For microbial inventories, a portion of feces and colon feces (discussed below) were frozen at −80 °C. All animals were maintained on a 3% oxalate diet, except during the washout period (*Supplementary file 1*).

Following diet trials, animals were sacrificed through $CO_2$ asphyxiation and cervical dislocation. Necropsies were performed to aseptically remove colon feces, colon, kidneys, and heart tissue. Colon tissue was longitudinally bisected, swiss-rolled, and placed into 4% paraformaldehyde overnight for fixation at 4 °C for histopathology (*Le Naour et al., 2022*). Kidney and heart tissues were placed into 4% paraformaldehyde overnight for fixation at 4 °C prior to histopathology. Hearts were first perfused with paraformaldehyde solution prior to immersion.

After fixation, colon, kidney, and heart tissue were placed in 70% ethanol at 4 °C prior to paraffin embedding, serial sectioning, and staining. To quantify colitis, colon tissue was stained with hematoxylin and eosin and imaged with light microscopy (*Figure 6H*). Whole colons were scored for colitis by two blinded, independent reviewers using a semi-quantitative protocol that considers inflammatory infiltrates, goblet cell loss, crypt density, crypt hyperplasia, muscle thickening, submucosal inflammation, crypt abscesses, and ulcerations (*Koelink et al., 2018*). To quantify calcium oxalate deposition, kidney tissue was stained with Von Kossa staining (*Rungby et al., 1993*), which turns calcium deposits black and the remaining tissue pink (*Figure 6E*, *Supplementary file 8E*). Kidneys were imaged under light microscopy and calcium oxalate deposition was quantified using an automated process in QuPath, based on stain color. The area of the black calcium oxalate deposits was normalized to the total kidney area. To quantify cardiac fibrosis, heart tissue was stained with Masson's trichrome stain, which stains collagen fibrosis blue and the remaining tissue red (*Figure 6—figure supplement 6*). Hearts were imaged under light microscopy and fibrosis was quantified with an automated process in QuPath, quantifying the blue and red stained areas (*Bankhead et al., 2017*). Fibrosis was normalized to total heart surface area.

## Biochemical assays

Urinary creatinine (Fisher Scientific), formate (VWR), IL-6 (R&D Systems), IL-18 (Fisher Scientific), and IL-1β (R&D Systems) were quantified with ELISA-based assays, following the manufacturer's instructions. Positive controls that included a known amount of substrate along with no substrate negative controls were included in all batches of ELISA assays. Samples, standards, and controls were all run in duplicate and values were averaged. Urinary and fecal oxalate were quantified with an ELISA-based assay (Sigma-Aldrich), with a modified protocol, is as follows. Upon initial testing of oxalate and formate assays using solutions of sodium oxalate and sodium formate in water, it was discovered that while formate assays only had affinity for formate, the oxalate assays had equal affinity for both oxalate and formate (*Figure 6—figure supplement 6A*). Specifically, when oxalate alone was added to the solution, the quantified concentration of oxalate matched the amount added. However, when oxalate and formate was added at equal concentrations, the quantified oxalate was approximately twice as high as what was added (*Figure 6—figure supplement 6B*). Furthermore, when adding 300 uM of formate or oxalate to human urine specimens, the amount of oxalate quantified was approximately 300 uM higher than urine without any modifications (*Figure 6—figure supplement 6C*). To eliminate the formate contamination on oxalate measurements, oxalate was extracted by first acidifying to pH 3 with 3 M $HNO_3$. Acidified urine was centrifuged, and supernatant collected. The pH was then brought up to 7 with NaOH. Subsequently, 5 M $CaCl_2$ was added to solution to precipitate calcium oxalate. The precipitates were extracted by centrifugation and decanting. Finally, calcium oxalate solution was acidified in 1:4 parts of 4 N HCl prior to enzymatic assay, as previously described (*Miller et al., 2016b*). When oxalate was extracted from urine specimens in this way, prior to quantification, we saw a significant decrease in quantified oxalate (*Figure 6—figure supplement 6C*), as expected given the assay affinity for formate. When solutions containing formate alone went through the extraction

process, no oxalate was quantified. Given these results, urinary and fecal oxalate was extracted prior to ELISA-based assays using the manufacturer's recommendations, post-extraction. For fecal oxalate, samples were acidified with 6 N $H_2SO_4$ to solubilize oxalate prior to following the extraction protocol above. For creatinine, cytokines, and oxalate, values generated on a 0% oxalate diet were substracted from all other values, matched by cage, with the assumption that this would isolate oxalate-induced molecule generation. For urinary formate, this substraction was not done since formate can come from multiple sources other than oxalate degradation and values on a 0% oxalate diet were not clearly lower than on a 3% oxalate diet. For all values, data are presented as the change from baseline. Data were statistically compared with two-way analyses and post-hoc, Holm's-corrected, paired t-tests.

## Culturomic assays

All assays were conducted under strict anaerobic conditions (90% $N_2$, 5% $H_2$, 5% $CO_2$) in an anaerobic chamber (Bactron 300).

The stool from *N. albigula* were used to determine the impact of environmental factors on oxalate metabolism in vitro and to isolate bacteria for the metabolic cohort. *Enterococcus gallinarum,* an oxalate-degrading species previously isolated from *N. albigula*, was also used in culturomic assays to determine the impact of environmental factors on oxalate metabolism (*Miller et al., 2014*). To test environmental factors on oxalate metabolism, gut microbiota media, previously designed to support a broad array of gut bacteria (*Goodman et al., 2011*), was used as a base media. From there, compounds were added to the media based on metabolic pathways enriched by dietary oxalate in the *N. albigula* gut microbiota (*Figure 3D*, *Supplementary file 6*). Concentrations of added chemicals were based on previously published media recipes where necessary. For assays using the NALB stool, the microbiota was extracted by first vortexing thawed stool, then centrifuging at 8000 rpm for 2 min. For in vitro assays involving pure cultures or synthetic cohorts, bacteria were first grown in media, with or without oxalate, that was first validated to allow for the growth of each isolate in pure culture (*Supplementary file 6*). Subsequently, extracted microbiota or pure cultures were normalized to $10^8$ microorganisms, as above. Pure cultures were mixed in equal proportions as needed and inoculated into the target media to evaluate oxalate metabolism under different chemical or microbial profiles. Media was incubated for 48–120 hr at 37 °C. All microbiological procedures were done under anaerobic conditions in a Bactron300 anaerobic chamber or sealed containers, as needed. To assess oxalate metabolism in different culture conditions, oxalate was quantified by extracting oxalate and quantification through titration with $KMnO_4$, as we have done previously (*Miller et al., 2014*). To assess the impact of oxalate on the growth of *E. gallinarum* or the NALB community, bacteria were grown in their respective media, with or without oxalate, and colony-forming units were quantified from serially diluted media after the incubation period.

To isolate bacteria for the metabolic cohort and to validate the metabolic redundancy of oxalate metabolism and other pathways stimulated by oxalate, a minimal media with the same substrates from above were added as sole carbon and energy sources (*Supplementary file 7*). In this medium, oxalate was added at 50 mM to specifically select for bacteria that can use oxalate as a carbon and energy source. Subsequently, the NALB microbiota from the stool was first diluted to ~1 bacterium/100 ul and then inoculated into 5 or 10 96-well plates, depending on the rarity of positive hits, and quantifying growth through spectrophotometry. Wells in which the absorbance values at 600 nm for greater than one standard deviation above the average for all wells were considered positive for growth. Wells with positive growth had bacteria isolated by streaking on rich media, transferred to broth cultures, and had stocks made in 15% glycerol that were frozen at –80 °C until needed. Bacteria were chosen for downstream in vitro and in vivo studies (metabolic cohort) by screening a minimum of ten isolates for each target substrate (*Figure 5D*). Those isolates grown in oxalate or formate-containing media were subsequently enriched in the minimal media and oxalate or formate degradation was validated through broth cultures and ELISA-based assays. Isolates that utilized the greatest amount of oxalate and formate were chosen for the metabolic cohort. For the other substrates, isolates were chosen based on those that exhibited the greatest amount of growth using the target substrates as a sole carbon and energy source, based on absorbance values at 600 nm. Subsequently, chosen isolates for the metabolic cohort along with isolates for the taxonomic cohort underwent PCR with the 27 F and 1492 R primers to amplify the 16 S rRNA gene. Amplicons were purified and sequenced with Sanger

sequencing to determine or validate taxonomy and purity. Taxonomy of isolates and the substrates each could utilize are presented in *Figure 5D*.

## Metabolomics

For untargeted metabolomic assays, fecal samples were re-suspended in Optima LC/MS grade water 50 mg stool to 150 µl water. The samples were then placed briefly in a water bath at 37 °C and placed on dry ice. An aliquot of each sample was taken out for protein concentration measurement. Then, 150 µL of chilled acetonitrile containing internal standards (Betaine-d9, Carnitine-d9, Orinithine-d6, Valine-13C3, Tyrosine-13C, 15 N, Estrone-13C3, and Cholesterol-13C3) was added to the remaining samples followed by centrifugation at 14,000 g for 10 min to precipitate out the protein pellet. The supernatants were recovered and transferred to fresh tubes. The samples were dried briefly and re-suspended in 2% acetonitrile and 0.1% formic acid for subsequent LCMS analysis. One-microliter aliquots taken from each sample were pooled and this QC standard was analyzed every 10 injections. The untargeted metabolomics was performed by injecting 0.5 uL of each sample onto a 10 cm C18 column (Thermo Fisher CA) coupled to a Vanquish UHPLC running at 0.35 mL/min using water and 0.1% formic acid as solvent A and acetonitrile and 0.1% formic acid as solvent B. The 15 min gradient used is given below. The Orbitrap Q Exactive HF was operated in positive and negative electrospray ionization modes in different LC-MS runs over a mass range of 50–750 Da using full MS at 120,000 resolution. The data-dependent acquisitions were obtained on the pooled QC sample. The DDA acquisition (DDA) includes MS full scans at a resolution of 120,000 and HCD MS/MS scans taken on the top 10 most abundant ions at a resolution of 30,000 with dynamic exclusion of 4.0 s and the apex trigger set at 2.0–4.0 s. The resolution of the MS2 scans were taken at a stepped NCE energy of 20.0, 30.0, and 45.0.

XCMS was used to deconvolute the data using 5 ppm consecutive scan error, 7–75 s as minimum and maximum peak width, S/N threshold of 10, and span of 0.2 in positive and negative electrospray ionization modes for retention time correction. The resulting peak table was normalized to total protein concentration and further analyzed via MetaboLyzer (*Mak et al., 2014*). Briefly, the ion presence threshold was set at 0.7 in each study group. Data were then log-transformed and analyzed for statistical significance via non-parametric Mann-Whitney U test. Ions present in just a subset of samples were to be analyzed as categorical variables for presence status via Fisher's exact test. All p-values were set to be corrected via the Benjamini-Hochberg step-up procedure for false discovery rate (FDR<0.05). The data could then be utilized for PCA, putative identification assignment, and pathway enrichment analysis via KEGG (*Okuda et al., 2008*), HMDB (*Wishart et al., 2018*), LIPID-MAPS (*O'Donnell et al., 2019*), and BioCyc (*Karp et al., 2019*) databases. In addition to differential abundance analysis provided by MetaboLyzer, whole metabolome comparisons were made as a Binomial dissimilarity matrix, which is optimal for metabolomic datasets (*Qi and Voit, 2017*), with statistical analysis provided by PERMANOVA with 999 permutations and a principal component visualization in Vegan (*Oksanen et al., 2010*).

## Transcriptomics

For transcriptomic analysis, RNA was extracted from liver tissue with a TRIzol RNA purification kit (Invitrogen), following the manufacturer's recommendations. Extracted RNA was converted to cDNA using a cDNA synthesis kit (Fisher Scientific) and stored at –20 °C. The cDNA quality was checked on a bioanalyzer and libraries were prepared with the TruSeq stranded mRNA library prep kit. Sequencing occurred on an Illumina HiSeq 2500 at the Genomics Core (Cleveland Clinic) for paired 100 bp sequencing.

The raw sequencing files, with 33.5 million+/-950,000 reads per sample, were pre-processed using an established protocol (*Delhomme et al., 2014*). Specifically, FastQC and SortMeRNA was used to remove low quality reads, remove rRNA reads, and merge paired reads (*Kopylova et al., 2012*; *Andrews et al., 2010*). Subsequently, trimmomatic was used to remove adapter sequences (*Bolger et al., 2014*) and the remaining high quality reads were mapped to the mm10 mouse reference genome for annotation, using HISAT (*Karolchik et al., 2014*; *Pertea et al., 2016*). The annotated gene counts were normalized to fragments per kilobase of transcript per million mapped reads. Comparative whole transcriptome analyses were conducted as a Bray-Curtis dissimilarity index and principal components visualization with statistical analysis provided by PERMANOVA with 999 permutations,

in the Vegan package of R statistical software (*Oksanen et al., 2010*; *R Development Core Team, 2013*). Differential abundance analysis was conducted in Cuffdiff2, with significance defined at a false discovery rate <0.05 (*Trapnell et al., 2013*). Sequences mapping to mouse genes were annotated to metabolic pathway with the gene ontology database (*Bult et al., 2019*).

To integrate transcriptomic and metabolomic data, the normalized metabolite concentrations and normalized transcript counts of metabolites and genes that were significantly different between the SWM-SWM and SWM-NALB groups were correlated, within each group, using SPARCC correlations in R (*Watts et al., 2019*). Significant positive and negative correlations were calculated at an R greater or less than 0.4 or –0.4, respectively and an FDR <0.05. Significant correlations were visualized in Cytoscape version 3.3.0 (*Shannon et al., 2003*).

## Shotgun metagenomics

For shotgun metagenomic analysis, DNA from *N. albigula* stool was extracted with the QIAmp DNA stool minikit (Qiagen). The extracted DNA was submitted to Argonne National Laboratory (Chicago, IL) for 150 bp paired-end shotgun metagenomic sequencing on an Illumina HiSeq 2500. The low quality reads were trimmed from raw sequencing data and paired ends were merged using default parameters in BBMerge (*Bushnell et al., 2017*). Reads were first mapped to the *Neotoma* genome (*Campbell et al., 2016*) with BWA mem with default settings (*Li, 2013*), and removed. Paired, non-rodent sequences were assembled in MetaSpades with default parameters (*Nurk et al., 2017*).

Full-length microbial genes were extracted and annotated from the assembled contigs using PROKKA software (*Seemann, 2014*). Gene annotation was achieved by mapping to the UniProt and Hamap protein databases (*Consortium, 2015*; *Pedruzzi et al., 2015*). To dereplicate genes, ensure common nomenclature across samples, and build a reference gene catalog for the quantification of gene counts, full-length genes were clustered at 90% homology in CD-hit (*Fu et al., 2012*). After the creation of a non-redundant gene catalog, high quality reads for each sample were mapped to the annotated gene catalog to generate gene-level count tables with BWA mem.

Gene count tables were normalized and differential abundance analysis was conducted with a negative binomial Wald test in DESeq2 (*Love et al., 2014*). Significantly different genes were considered at an FDR <0.05. The normalized count tables were also used to generate a weighted Bray–Curtis dissimilarity matrix in the Vegan R package, with statistical comparisons with a PERMANOVA at 999 permutations (*Oksanen et al., 2010*). To annotate differentially abundant genes to taxonomy, genes were mapped to full-length genomes extracted directly from the shotgun metagenomic data, as described below.

## de novo genome construction and calculation of metabolic redundancy

For the de novo construction of genomes, the metaspades assembled contigs were binned to genomes with Autometa software (*Miller et al., 2019c*), as previously described (*Kachroo et al., 2022*). The Autometa algorithm takes assembled contigs as input, bins the data to genomes using contig coverage values, GC content, and the Barnes-Hut t-Distributed Stochastic Neighbor Embedding (BH t-SNE) distribution of the contigs. Taxonomy is assigned based on the consensus classification of all contigs in a genomic bin. Completeness and purity calculations are based on the presence of known, unique universal single-copy genes in Autometa. After genomic binning, the contigs were oriented and scaffolded in CSAR (*Chen et al., 2018*), which compares the genomic bins to reference genomes of close relatives. Gaps within the genomes were filled with Abyss-Sealer (*Simpson et al., 2009*) using the original raw sequencing reads as input. The completeness and purity of genomes were recalculated and taxonomic assignment was validated through phylogenetic analysis of the de novo constructed genomes in comparison with all complete NCBI genomes (*Supplementary file 1*), achieved through Phylophlan, using the '—accurate' parameter which considers more phylogenetic positions at the cost of computational speed (*Asnicar et al., 2020*). Phylophlan allows for the phylogenetic analysis of bacteria and archaea using complete genomes rather than gene amplicons, which provides greater resolution on phylogenetic analysis. Collectively, we extracted 248 high quality, full-length genomes, defined as being >80% complete and >90% pure, based on current standards (*Bowers et al., 2017*).

To quantify the proportion of the *N. albigula* microbiota that contained genes related to oxalate metabolism, formate metabolism, acetogenesis, methanogenesis, or sulfate reduction, all genes that

contribute to those metabolic pathways were extracted from the KEGG database (*Kanehisa et al., 2019*). Subsequently, PROKKA (*Seemann, 2014*) was run on the 248 full-length genomes to extract and annotate full-length genes. The PROKKA extracted genes were then cross-referenced with the gene databases from KEGG to determine the proportion of genomes that contained at least one gene in the target pathways and determine which taxa had those genes.

## 16S rRNA metagenomics

Fecal DNA from the stool and colon feces from the mouse studies, underwent DNA extraction through a semi-automated protocol on a KingFisher Duo Prime System (Thermo Scientific) following the manufacturer's protocol for stool. The protocol includes mechanical lysis with bead-beating, piston driven lysis, and proteinase K chemical lysis. The extracted DNA, which attaches to magnetic beads in a proprietary salt solution, is removed from solution and eluted in buffer. Duplicate samples of a commercial DNA positive of known composition (Zymboiomcs, USA) were included as positive controls, as well as the DNA reagents that went through the entire workflow, sterile water, and a PCR negative as negative controls.

The fecal DNA was submitted to the Microbial Sequencing and Analytics Core at Cleveland Clinic for high throughput sequencing of the 16 S rRNA gene on an Illumina MiSeq. The DNA was first PCR amplified with the 515 F and 806 R primers that target the V4 hypervariable region of the 16 S gene. The DNA concentrations were quantified before and after PCR amplification on a Qubit and normalized prior to library prep with the Illumina Nextera XT library prep kit. The sequence run was conducted to generate 150 bp, paired-end sequences.

The raw sequencing data were processed in the R statistical package (4.2.0) (*R Development Core Team, 2013*). Quality control, removal of bimeras, and assignment of amplicon sequence variants (ASV) were completed in dada2 (*Callahan et al., 2016*). For ASV taxonomic assignment, a combined, non-redundant database of the Silva 138 SSURef and NCBI 16 S rRNA databases were used (*Balvočiūtė and Huson, 2017*). Taxa assigned to mitochondria or chloroplasts were removed from subsequent analysis. The resulting ASV's were aligned in MSA (*Bodenhofer et al., 2015*) and arranged into a maximum likelihood phylogeny in phangorn (*Schliep, 2011*). The resulting phylogenetic tree and ASV table were merged with sample data for loading into PhyloSeq (*McMurdie and Holmes, 2013*). The sequencing depth threshold required to adequately capture microbial diversity was calculated with a rarefaction analysis in Vegan (*Oksanen et al., 2010*). Non-control samples below the depth threshold were removed from further analysis.

The raw count table of 16 S rRNA sequences was normalized with the DESeq2 algorithm and α-diversity was calculated as phylogenetic diversity in Phyloseq, balong with beta-diversity as a weighted UniFrac distance (*Lozupone et al., 2006*). Alpha diversity was analyzed with a paired t-test with a Holm's correction where applicable, while beta-diversity analyses were conducted with a PERMANOVA after 999 permutations.

## Acknowledgements

We would like to thank the animal care facility, Metabolomics core, Microbial sequencing and analytics core, and Imaging core at Cleveland Clinic. We are thankful for the NIH funding to AWM R01 DK121689-01A1 and funding from the Cleveland Clinic Foundation to AWM.

## Additional information

### Funding

| Funder | Grant reference number | Author |
|---|---|---|
| National Institutes of Health | DK121689-01A1 | Aaron W Miller |
| Cleveland Clinic Foundation | | Aaron W Miller |

| Funder | Grant reference number | Author |
| --- | --- | --- |

The funders had no role in study design, data collection and interpretation, or the decision to submit the work for publication.

## Author contributions

Sromona D Mukherjee, Conceptualization, Data curation, Formal analysis, Investigation, Methodology, Writing – original draft, Writing – review and editing; Carlos Batagello, Jose Agudelo, Andrew Nguyen, Conceptualization, Formal analysis, Investigation, Methodology, Writing – review and editing; Ava Adler, Data curation, Investigation, Methodology, Writing – review and editing; Anna Zampini, Conceptualization, Data curation, Formal analysis, Writing – review and editing; Mangesh Suryavanshi, Conceptualization, Data curation, Formal analysis, Investigation, Writing – review and editing; Terry Orr, Denise Dearing, Investigation, Methodology, Writing – review and editing; Manoj Monga, Conceptualization, Investigation, Methodology, Writing – review and editing; Aaron W Miller, Conceptualization, Data curation, Formal analysis, Supervision, Funding acquisition, Validation, Investigation, Methodology, Writing – original draft, Project administration, Writing – review and editing

## Author ORCIDs

Sromona D Mukherjee https://orcid.org/0000-0003-1958-0220
Aaron W Miller https://orcid.org/0000-0001-8342-1449

## Ethics

This study was performed in strict accordance with the recommendations in the Guide for the Care and Use of Laboratory Animals of the National Institutes of Health. All of the animals were handled according to approved institutional animal care and use committee (IACUC) protocols (#12-12010) of the University of Utah and (#2016-1653) of Cleveland Clinic.

Reviewer #2 (Public review): https://doi.org/10.7554/eLife.104121.3.sa1
Author response https://doi.org/10.7554/eLife.104121.3.sa2

---

# Additional files

## Supplementary files

Supplementary file 1. Diets used in animal studies.

Supplementary file 2. The number of host hepatic genes in each metabolic pathway that exhibited a significant increase (positive numbers) or decrease (negative numbers) in expression upon consumption of a 1.5% oxalate diet.

Supplementary file 3. The number of microbial metabolites in each metabolic pathway that exhibited a significant increase (positive numbers) or decrease (negative numbers) in expression upon consumption of a 1.5% oxalate diet by the host.

Supplementary file 4. The significant positive and negative correlations between microbial metabolites and host hepatic gene expression for those metabolites/genes significantly altered by 1.5% oxalate consumption. Data are listed with the host microbiome, metabollite and gene ID, long with correlation values, and false discovery rate corrected p-values.

Supplementary file 5. The number of microbial genes in each metabolic pathway that exhibited a significant increase (positive numbers) or decrease (negative numbers) in abundance upon consumption of a 6% oxalate diet by the host.

Supplementary file 6. Media recipe used to test the impact of substrates on oxalate metabolism.

Supplementary file 7. Media recipe used to test the proportion of the *N. albigula* (NALB) community that can utilize substrates as sole carbon and energy sources.

Supplementary file 8. Microorganisms used in the taxonomic cohort.

Supplementary file 9. Information for 248 complete genomes extracted from the shotgun metagenomic data from the *N. albigula* gut microbiota.

MDAR checklist

## Data availability

Sequence reads from the animal study are available at the Sequence Read Archive under Accession numbers for gut microbiota genomes: PRJNA833303, Shotgun metagenomics: PRJNA839366, Liver transcriptomics: PRJNA1018952, Gut microbiota data: 16S PRJNA1018559. Source data and code is provided on GitHub (copy archived at *Miller, 2025*).

The following datasets were generated:

| Author(s) | Year | Dataset title | Dataset URL | Database and Identifier |
|---|---|---|---|---|
| Cleveland Clinic | 2022 | Gut microbiota of wild rodents (Neotoma albigula) on a no or high oxalate diet | https://www.ncbi.nlm.nih.gov/bioproject/PRJNA833303/ | NCBI BioProject, PRJNA833303 |
| Cleveland Clinic | 2022 | Neotoma albigula | https://www.ncbi.nlm.nih.gov/bioproject/?term=PRJNA839366 | NCBI BioProject, PRJNA839366 |
| Cleveland Clinic | 2023 | Effect of oxalate on liver transcriptome | https://www.ncbi.nlm.nih.gov/bioproject/?term=PRJNA1018952 | NCBI BioProject, PRJNA1018952 |
| Cleveland Clinic | 2023 | Impact of oxalate-degrading isolates from woodrats on hyperoxaluria in mouse model | https://www.ncbi.nlm.nih.gov/bioproject/?term=PRJNA1018559 | NCBI BioProject, PRJNA1018559 |

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
