## [Editor Report · eLife Assessment]

This work presents a **valuable** approach based on a complex systems theoretical framework to characterize diet-host-microbe interactions and develop targeted bacteriotherapies through a three-phase workflow. Despite the partial support of the description and experimental setup of the 'complex systems theoretical approach,' the collected data are **solid** and advance our understanding of oxalate bacterial metabolism in microbial communities. This study will interest researchers working on gut microbiomes and the possible modulation of host-microbial interactions.

---

## [Referee Report · Reviewer #2 (Public review)]

Summary:

Using the well-studied oxalate-microbiome-host system, the authors propose a novel conceptual and experimental framework for developing targeted bacteriotherapies using a three-phase pre-clinical workflow. The third phase is based on a 'complex system theoretical approach' in which multi-omics technologies are combined in independent in vivo and in vitro models to successfully identify the most pertinent variables that influence specific phenotypes in diet-host-microbe systems. The innovation relies on the third phase since phase I and phase II are the dominant approaches everyone in the microbiome field uses.

Strengths:

The authors used a multidisciplinary approach which included [i] fecal transplant of two distinct microbial communities into Swiss-Webster mice (SWM) to characterize the host response (hepatic response-transcriptomics) and microbial activity (untargeted metabolomics of the stool samples) to different oxalate concentrations; [2] longitudinal analysis of the N. albigulia gut microbiome composition in response to varying concentrations of oxalate by shotgun metagenomics, with deep bioinformatic analyses of the genomes assembled; and [3] development of synthetic microbial communities around oxalate metabolisms and evaluation of these communities' activity into oxalate degradation in vivo.

Weaknesses:

This study presents a valuable finding on the oxalate-microbiome-host system using a multitude of approaches. Although the multidisciplinary approach allows for a unique perspective on the system and more robust conclusions, it is challenging for any authors to present all the data clearly and systematically in a conclusive way-especially when introducing unfamiliar concepts such as a complex systems theoretical approach.

---

## [Author Response]

The following is the authors’ response to the original reviews

**Reviewer #1 (Public review):**
Summary:This study experimentally examined diet-microbe-host interactions through a complex systems framework, centered on dietary oxalate. Multiple, independent molecular, animal, and in vitro experimental models were introduced into this research. The authors found that microbiome composition influenced multiple oxalate-microbe-host interfaces. Oxalobacter formigenes were only effective against a poor oxalate-degrading microbiota background and give critical new insights into why clinical intervention trials with this species exhibit variable outcomes. Data suggest that, while heterogeneity in the microbiome impacts multiple diet-host-microbe interfaces, metabolic redundancy among diverse microorganisms in specific diet-microbe axes is a critical variable that may impact the efficacy of bacteriotherapies, which can help guide patient and probiotic selection criteria in probiotic clinical trials.

Thank you. The main message of this research, is that through complex modelling, we believe we have identified the critical variable (metabolic redundancy) that is responsible for the efficacy of probiotics designed to reduce oxalate levels, thus allowing for improved patient selection in clinical trials. We also believe that this process and the critical features identified can be translated to other critical microbial functions such as short chain fatty acid synthesis, secondary bile acid synthesis, and others.

Strengths:The paper has made significant progress in both the depth and breadth of scientific research by systematically comparing multiple experimental methods across multiple dimensions. Particularly through in-depth analysis from the enzymatic perspective, it has not only successfully identified several key strains and redundant genes, which is of great significance for understanding the functions of enzymes, the characteristics of strains, and the mechanisms of genes in microbial communities, but also provided a valuable reference for subsequent experimental design and theoretical research.More importantly, the establishment of a novel research approach to probiotics and gut microbiota in this paper represents a major contribution to the current research field. The proposal of this new approach not only breaks through the limitations of traditional research but also offers new perspectives and strategies for the screening, optimization of probiotics, and the regulation of gut microbiota balance. This holds potential significant value for improving human health and the prevention and treatment of related diseases.

Thank you for the comments. We believe that the approach taken here, which contrasts with conventional reductionist techniques, will be critical for translating gut microbiome research into actionable therapeutic approaches.

Weaknesses:While the study has excellently examined the overall changes in microbial community structure and the functions of individual bacteria, it lacks a focused investigation on the metabolic cross-feeding relationships between oxalate-degrading bacteria and related microorganisms, failing to provide a foundational microbial community or model for future research. Although this paper conducts a detailed study on oxalate metabolism, it would be beneficial to visually present the enrichment of different microbial community structures in metabolic pathways using graphical models.

Thank you for this critique. In the current study, we broadly examined the response of the gut microbiota to dietary oxalate. Based on initial shotgun metagenomic results, we focused in on specific taxa and metabolic functions. Through metagenomic and multiple culture-based studies, we quickly honed in on redundancy in oxalate-degrading function as a key feature for oxalate homeostasis. We believe that the defined microbial community we used for microbial transplants (particularly the taxonomic cohort) provides a strong, minimal community to explore oxalate homeostasis further. In fact, we are using this consortium in multiple follow-up studies to fully understand the cross-feeding that may occur among these microorganisms, as you suggest. We note that figure 3 shows the change of species and metabolic pathways with oxalate exposure.

Furthermore, the authors have done a commendable job in studying the roles of key bacteria. If the interactions and effects of upstream and downstream metabolically related bacteria could be integrated, it would provide readers with even more meaningful information. By illustrating how these bacteria interact within the metabolic network, readers can gain a deeper understanding of the complex ecological and functional relationships within microbial communities. Such an integrated approach would not only enhance the scientific value of the study but also facilitate future research in this area.

Thank you. We note that based on the collective data obtained in this study, that redundancy in the oxalate degradation is the critical feature that maintains oxalate homeostasis. However, we are interested potential metabolic interactions between microbes in our defined community and are currently investigating these interactions through extensive investigations.

**Reviewer #2 (Public review):**
Summary:Using the well-studied oxalate-microbiome-host system, the authors propose a novel conceptual and experimental framework for developing targeted bacteriotherapies using a three-phase pre-clinical workflow. The third phase is based on a 'complex system theoretical approach' in which multi-omics technologies are combined in independent in vivo and in vitro models to successfully identify the most pertinent variables that influence specific phenotypes in diet-host-microbe systems. The innovation relies on the third phase since phase I and phase II are the dominant approaches everyone in the microbiome field uses.

Thank you. As you note, the proposed phases I and II are the predominant approaches used. In fact, many clinical trials have been conducted to try and reduce urine oxalate in patients, based solely on mechanistic studies with Oxalobacter formigenes. As noted in our manuscript, only 43% of those studies results in the intended outcome, necessitating the approach we took in the current study. Our results suggest that the reason for the high rate of failure, despite well established mechanisms, is due to insufficient patient selection that focused only on the presence or absence of *O. formigenes*, which is a species that exhibits very low prevalence and abundance in the human gut microbiota, normally.

Strengths:The authors used a multidisciplinary approach which included:(1) fecal transplant of two distinct microbial communities into Swiss-Webster mice (SWM) to characterize the host response (hepatic response-transcriptomics) and microbial activity (untargeted metabolomics of the stool samples) to different oxalate concentrations;(2) longitudinal analysis of the N. albigulia gut microbiome composition in response to varying concentrations of oxalate by shotgun metagenomics, with deep bioinformatic analyses of the genomes assembled; and(3) development of synthetic microbial communities around oxalate metabolisms and evaluation of these communities' activity in oxalate degradation in vivo.

Thank you for these comments. In the complex modelling approach, we focused on complete microbiota from host species known to have high and low capacities for oxalate tolerance, combined with targeting specific metabolic functions vs. specific taxa that may include unknown functions important for oxalate metabolism. Further, we examined the influence of our target communities on oxalate metabolism through multiple in vitro and in vivo studies.

Weaknesses:However, I have concerns about the frame the authors tried to provide for a 'complex system theoretical approach' and how the data are interpreted within this frame. Several of the conclusions the authors provide do not seem to have sufficient data to support them.

Thank you. We have tried to address these concerns by adding an exhaustive figure that broadly represents our complex modelling approach that includes potential complex system-based hypotheses, how they were tested, and the host-microbiome-oxalate interactions found in our study.

**Recommendations for the authors:**

**Reviewer #2 (Recommendations for the authors):**
Major Concerns(1) The authors argue about the importance of bringing 'Complex System Theory' to the microbiome field systematically and consistently. However, the authors fail to introduce this theory throughout the entire manuscript. For example, the authors tried to describe key elements and their nomenclature, such as nodes and fractal layers, in the first part of the result section. But the description is wordy and not precise. It would be more useful if the authors connected the model description with a visual representation, such as a figure. Unfortunately, these elements are not emphasizing and carried across the results section and are not mentioned in the discussion section.

We have now added a figure (Figure 7) that details this process extensively and ties each of our findings to the complex system model and nomenclature. We have also reiterated how our results fit in the complex system model in the discussion.

In addition, there is no straightforward approach to integrating multi-omics datasets to identify the variables that are determinants of the system. For example, Figure 1 focuses on the impact of the host, hepatic activity, to oxalate exposure on fecal transplants into Swiss Webster mice; Figure 2 focuses on the effects of oxalate exposure on stool metabolic activity, not only microbial metabolic activity, on fecal transplants into Swiss Webster mice; and Figure 3 focuses on microbiome responses to different oxalate concentration in Neotoma albigula. There is no "model" to really integrate the host, the microbiome activity, and the microbiome composition information. And, unfortunately, the data generated between experiments cannot directly integrate; see major concern # 2.

Thank you. We have made more clear the experimental approach and how it applied to understanding the critical factors that maintain oxalate homeostasis. Specifically, Figure 1 established that the effect of oxalate on the host was dependent on the microbiota, rather than host genetics. Figure 2 established the effect of oxalate on the gut microbiota was again dependent on the whole gut microbiota and that these oxalate-microbe effects also influenced oxalate-host effects through a direct multi-omic data integration. Once we established that the oxalate effects on host and microbiota were dependent on the whole microbiota composition, Figure 3 then sought to figure out how oxalate impacted the gut microbiota, using our model of high oxalate tolerance (*N. albigula*). With the finding in Figure 3 that there were multiple genes attributed to the degradation of oxalate, or acetogenic, methanogenic, and sulfate reducing pathways, Figure 4 and relevant supplemental figures sought to quantify the redundancy of these pathways. After establishing a very high degree of redundancy, we sought to use a culturomic approach to determine what environmental factors impacted oxalate metabolism and to evaluate oxalate metabolism using our defined, hypothesized communities of microorganisms. Finally, figure 6 sought to validate our metagenomic, metabolomic, and culturomic results from multiple animal and in vitro models using targeted microbial transplants in mice. While we did have some direct multi-omic data integration (Figures 2 and 3), the process employed here sought to systematically determine which factors were most important for the oxalate-microbiota-host relationship, and then to use those results to design the subsequent experiments. We have added this description to the discussion, which helps to contextualize the complex system modelling approach we took here.

Finally, the authors did not provide a novel variable that successfully influences oxalate degradation in the oxalate-microbiome-host system. The authors argue that "both resource availability and community composition impact oxalate metabolism," which we currently inferred by the failure of the clinical tries and do not provide a clear intervention strategy to develop functional bacteriotherapy. The identification of composition as an important variable that was predictable without any multi-omics approach was highlighted by the development of synthetic microbial communities. Synthetic microbial communities are critical to characterizing complex microbiomes. Still, the authors did not explain how this strategy can be used in their theoretical framework (that is their goal), and these communities are not well introduced across the manuscript; see major concern # 4.

As stated, it is clear from the failed clinical trials that we do not fully understand what microbial features dictate oxalate homeostasis. We have specifically identified, through fecal transplant studies, that microbial composition is critical for oxalate homeostasis and that diverse oxalate-degrading bacteria exist. However, ours is the first study that explicitly shows that it is this diversity that controls oxalate homeostasis. This is specifically ascertained through the targeted microbial transplants in mice whereby *O. formigenes* was given alone or with different combinations of other microorganisms. In other words, we were able to replicate both successful and failed studies by manipulating which specific species were introduced into animals. This is unprecedented in the literature.

(2) The authors provide several conclusions that are not completely supported by the data available. For example:(a) Lines 236-239: "Within the framework of complex systems, results show microbe-host cooperation whereby oxalate effectively processed within the SW-NALB gut microbiota reduced overall liver activity, indicative of a beneficial impact." - The authors did not provide data related to oxalate levels of oxalate processing for this dataset.

While we did not specifically quantify oxalate degradation for this specific study, as cited in the text when describing this Swiss-Webster, Neotoma albigula system, we have previously published multiple animal studies explicitly showing that the *N. albigula* animals were highly effective oxalate degraders, which is transferable to Swiss-Webster mice through fecal transplants. Since the gut microbiota’s impact on oxalate has been welll established through experiments by our group, the purpose of these specific experiments were to look the other way and examine the effect of oxalate on the gut microbiota of these two animal models. In the referenced text, we again cited our studies showing that the SW-NALB system effectively degrades oxalate.

(b) Lines 239-243: "Data also suggest that both the gut microbiota and the immune system are involved in oxalate remediation (redundancy), such that if oxalate cannot be neutralized in the gut microbiota or liver, then the molecule will be processed through host immune response mechanisms (fractality), in this case indicated through an overall increase in hepatic activity and specifically in mitochondrial activity." - The authors did not provide any evidence related to the immune system and oxalate metabolism.

We corrected that statement as follows: “…in this case indicated through an overall increase in inflammatory cytokines with oxalate exposure combined with an ineffective oxalate-degrading microbiota (Figures S6a,b; S9a,b).” In other words, if the liver and gut microbiota can’t eliminate a toxin, then the immune system must deal with it through inflammatory pathways. Oxalate is a well established, pro-inflammatory compound. Our data show that this is dependent on the gut microbiota.

(c) Lines 250-252: "Following the diet trial, colon stool was collected post-necropsy and processed for untargeted metabolomics, which is a measure of total microbial metabolic output." - Although most metabolites in stool samples are indeed microbial, there are also host metabolites. So, it is not technically correct to relate the metabolomic analysis of stool samples to only microbial metabolic analysis. In addition, the authors discussed compounds such as alkaloids and cholesterol as microbial metabolites, which these compounds are more related to the diet and host correspondingly.

We have corrected this to state: “total metabolites present in stool from the diet, microbial activity, and host activity”

(d) Lines 270-273. "Specifically, the SW-NALB mice exhibit hallmarks of homeostatic feedback with oxalate exposure to maintain a consistent metabolic output, defined by the relatively small, net negative, microbial metabolite-hepatic gene network compared to the large, net positive, network of SW-SW mice." - How do the authors define oxalate homeostasis? In addition, do the authors imply feedback between the liver and the microbiome in which the microbiome responds to a liver response related to oxalate levels? Or could the observation in Figure 1 be explained just by microbial consumption of oxalate that would reduce the impact of oxalate that arrives at the liver?

Oxalate homeostasis is defined in that sentence: “relatively small, net negative, microbial metabolite-hepatic gene network compared to the large, net positive, network of SW-SW mice” – in other words, for SW-NALB mice, oxalate did not produce a considerable change to either microbial or hepatic metabolic activity. We did not really test the liver impact on gut microbiota and can’t speak to that. We believe, based on Figure 2 data, that it is not just the degradation of oxalate that explains the lack of change in hepatic activity in SW-NALB mice, rather that the oxalate-induced shift in the gut microbiota metabolic activity broadly altered hepatic activity, as inferred from Figure 2 c. We made this more clear in the results: “suggests that the oxalate-induced change in microbial metabolism is responsible for the change in hepatic activity”.

(e) Lines 297-301: "The oxalate-dependent metagenomic divergence of the NALB gut microbiota (Figure 3), combined with the lack of change in the microbial metabolomic profile with oxalate exposure (Figure 2), suggest that oxalate stimulates taxonomically diverse, but metabolically redundant microorganisms, in support of maintaining homeostasis." - The authors cannot conclude anything related between taxonomic changes and microbial activity since the taxonomic data presented is for microbial enrichment in N. albigulia, and the "microbial activity data" is from the fecal transplantation experiment in SWM. These are two completely different systems with two completely different experimental designs.

We have shown very similar results in that oxalate induces the taxonomic divergence for the NALB gut microbiota, in multiple previous studies. The experiment in which a minimal, positive increase in microbial metabolites, was saw with oxalate was based on the SW-NALB model whereby Swiss-Webster mice have an NALB microbiota. We show throughout the manuscript, that the impact of oxalate is very microbiota dependent and supports our claim. However, the claim is hypothesis generating – that metabolic redundancy is important for oxalate homeostasis. We modified our statement to make all of this more clear.

Related to microbial composition, the authors did not show data validating the efficiency of the fecal transplantations (allograft or xenograft) in the SWM after antibiotic treatment. They also did not show evidence of microbial composition dynamics in response to oxalate exposure.

Again, the efficacy of fecal transplants, used in the way they were here, has been shown in multiple past studies of our group. In past studies, we have extensively characterized the microbiota from fecal transplants and which taxa were associated with oxalate levels. Therefore, that topic was not the focus of the current study, instead focusing on the oxalate impact on gut microbiota activity. Our past studies, referenced multiple times through the current manuscript, were used in large part to help determine which microbes to include in our taxonomic cohort, as described in the manuscript.

(f) Lines 301-303: "Given that data came from the same hosts sampled longitudinally, these data also reflect a microbiota that is adaptive to oxalate exposure, which is another important characteristic of complex systems." - In their dataset, what is the evidence that the microbiota of N. albigulia is adapted to oxalate exposure? Is the increase in genomes with pathways related to oxalate metabolism related to an increase of oxalate in the diet? If so, does the microbiota exposure with a higher oxalate concentration decrease the systemic level of oxalate? In neither of the experiments related to Figures 1 to 3, the authors showed a correlation of systemic oxalate levels with microbial composition, hepatic host response, or stool metabolism.

Figure 3 explicitly shows the longitudinal impact of increasing levels of oxalate showing an increase in oxalate degrading genes (Figure 3d). The specific samples selected for analysis here come from a previous study in which we explicitly quantified changes to the gut microbiota composition and both stool and urine oxalate for every time point listed in figure 3a. This information is explicitly stated in the methods coupled with the fact that “neither fecal nor urinary oxalate levels increased significantly.” Again, the effect of the gut microbiota on oxalate in these model systems have been extensively studied by our group and provide the foundation for the current study to look at the effect of oxalate on the gut microbiota and host.

Considering my last two points, the authors do not present substantial evidence to support their hypothesis that oxalate stimulates taxonomically diverse, metabolically redundant communities.

As stated above, that oxalate stimulates taxonomically diverse taxa was ascertained through multiple past studies, as well as the current study (Figure 3e). The metabolically redundant part is ascertained both through untargeted metabolomics (Figure 2a,b) and shotgun metagenomics (Figure 3c,d). Further evidence for the metabolic redundancy with oxalate comes from our culturomic approach, which showed that 14.58% of isolates could grow on oxalate as a carbon and energy source, in addition to the high proportion of isolates that could grow on other carbon and energy sources, at least much more than can be ascribed to a single species (Figure 5c). We made this more clear in the discussion.

(g) Lines 330-335. "Additionally, the broad diversity of species that contain oxalate-related genes suggests that the distribution of metabolic genes is somewhat independent of the distribution of microbial species, which suggests that microbial genes exist in an autonomous fractal layer, to some degree. This hypothesis is supported by studies which show a high degree of horizontal gene transfer within the gut microbiota as a means of adaptation." - This conclusion is highly speculative, especially since the author did not do any analysis to directly evaluate a relationship between the oxalate metabolic pathways and the microbial species where these pathways are present.

Figure 3c,d,e explicitly shows the metabolic pathways and species enriched by oxalate exposure. Figure 4d, generated using the same data from Figure 3, explicitly shows the taxa that harbor oxalate-degrading genes.

(h) Lines 364-366. "Collectively, data show that both resource availability and community composition impacts oxalate metabolism, which helps to define the adaptive nature of the NALB gut microbiota." - The authors indeed showed evidence that community composition impacts oxalate metabolism. However, the authors did not show any evidence to directly evaluate the resource availability to impact oxalate metabolism.

This is explicitly shown through in vitro community-based and single species assays varying multiple different carbon and energy sources to quantify changes to oxalate degradation (chosen based on shotgun metagenomic results; Figure 5a,b).

(3) Lines 321-325. "Acetogenic genes were also present in 97.18% of genomes, dominated by acetate kinase and formate-tetrahydrofolate ligase (Figure S3A323C). Methanogenic genes were present in 100% of genomes, dominated by phosphoserine phosphatase, atpdependent 6-phosphofructokinase, and phosphate acetyltransferase (Figure S4A-C)." - The authors spent much time analyzing the adjacent pathways related to oxalate and oxalaterelated products of oxalate metabolism. However, my understanding is that the genes used to analyze these pathways (formate metabolism, acetogenesis, methanogenesis), such as the ones named above, are not unique/specific for those pathways but participate in other "housekeeping" pathways. What is the relevance of these analyses when those genes are not unique/specific to the function/pathways that the authors describe? If I infer correctly, these bioinformatic analyses aim to evaluate the hypothesis of whether oxalate metabolism could be a social/cooperation metabolism and whether other species could participate in the metabolism of oxalate subproducts. However, these analyses did not explicitly evaluate this hypothesis.

The reviewer is correct in that we aimed to evaluate the potential that oxalate metabolism could benefit from metabolic cooperation. The specific genes chosen for this analysis were those explicitly listed in the target metabolic pathways in KEGG, as described. However, while the analyses do show the strong potential that the CO2 and formate produced from oxalate degradation could be used in these other pathways, as intended, the genes can be used in other metabolic pathways. We did, however, explicitly test the hypothesis that formate, produced from oxalate degradation, could be utilized by the gut microbiota. While the targeted transplants with the taxonomic cohort did not clearly show the use of formate in this way, those from the metabolic cohort did (Figures 6d and S8d). This question is still in ongoing investigations in our group.

We have made it more clear that our genome analyses provide the potential for metabolic redundancy rather than definitive proof for metabolic redundancy, which was evaluated more extensively in other experiments from this study.

(a) Lines 481-484. "Collectively, data offer strong support for the hypothesis that metabolic redundancy among diverse taxa, is the primary driver of oxalate homeostasis, rather than metabolic cooperation in which the by-products of oxalate degradation are used in downstream pathways such as acetogenesis, methanogenesis, and sulfate reduction." - Although the authors recognize that their data about the metabolic cooperation hypothesis is inconclusive, they never tested the hypothesis related to metabolic cooperation, as mentioned above. This is highly speculative.

As stated above, the targeted microbial transplants to animals and in vitro studies (Figure 5e,f) did explicitly test the cooperation hypothesis, but it the results did not support it and instead pointed much more strongly to metabolic redundancy.

(4) Lines 355-359. "Cohorts, defined in the STAR methods, were used to delineate hypotheses that either carbon and energy substrates are sufficient to explain known effects of the oxalate-degrading microbial network or that additional aspects of taxa commonly stimulated by dietary oxalate are required to explain past results (taxa defined through previous meta-analysis of studies)." - The definition of the metabolic cohorts and the taxonomic cohorts should not be hidden in the material and methods section. It should be explicit and clearly explained in the main text. Related, the table presented in Figure 5D is exceptionally confusing and does not help to understand and differentiate between the metabolic and the taxonomic cohorts. The authors need to explicitly identify the synthetic communities used in each cohort and each group by their members and their characteristics in supplementary tables.

In the sentences before those referenced, we state: “Culturomic data recapitulates molecular data to show a considerable amount of redundancy surrounding oxalate metabolism (Fig. 5C). Isolates generated from this assay were used for subsequent study (metabolic cohort; Figure 5D). Additionally, a second cohort was defined and commercially purchased based both on known metabolic functions and the proportion of studies that saw an increase in their taxonomic population with oxalate consumption (Fig. 5D; taxonomic cohort). Where possible, isolates from human sources were obtained.” Figure 5d explicitly shows the specific species used in each cohort along with the groups they were in for transplant studies, the explicit metabolic pathways we were targeting, along with the % of studies that these species were associated with oxalate metabolism. All of this information is both in the main text of the results and in the figure legends. It is not hidden in the methods, but the methods do reiterate what was also placed in the results.

In Figures 5 and 6, the authors used the following groups with the corresponding nomenclature: 'Group 1, No_bact; Group 2, Ox; Group 3, Ox_form; Group 4, All; Group 5, No_ox'. Although the information related to these groups is present in the material and method section in lines 1139-1143, the authors also need to explicitly explain the groups and their nomenclature in the main text.

Since this information is explicitly and succinctly given in the referenced figures, I believe that adding the same information in the text would be too redundant.

Related to the development of the synthetic communities. How did the authors prepare the synthetic communities or 'cohort' for the in vitro experiments?

We added more information for the preparation of microbes and execution of the in vitro assays, as needed.

Also, it is unclear in the material and method section how the metabolic profile of each isolated was evaluated (Figure 5C). Related to the bacteria isolated from the culturomic assays, including Figure 5C and metabolic cohort, the authors indeed reported the isolation methodology in lines 1262-1275. However, there is no information about the sequencing of these isolates. The authors should present these isolates as a list (supplementary table) with their names, taxonomy, metabolic profile, and Genome ID if these genomes were submitted to NCBI.

We added additional information for how metabolic cohort isolates were chosen and how they were taxonomically identified. The taxonomy and substrate utilization of isolates are in Figure 5D. We did not sequence the genomes of metabolic cohort bacteria. However, the ATCC isolates, which comprise the taxonomic cohort, are publicly available.

The author presented the 248 metagenomics assembles in Figure S1 in a circular chart in context with other genomes. However, the metagenomic assembles should be presented in a table form, with their name, taxonomy, coverage, completeness, and Genome ID, if these genomes were submitted to NCBI.

The information for the genomes submitted to the NCBI is provided in the data availability statement. However, we added a table (Table S9) that includes the requested information.

(5) Lines 371-3374: "To delineate hypotheses of metabolic redundancy or cooperation for mitigating the negative effects of oxalate on the gut microbiota and host, two independent diet trials were conducted with analogous microbial communities derived from the metabolic and taxonomic cohorts".Lines 494-496: "we and others have found that oxalate can differentially exhibit positive or negative effects on microbial growth and metabolism dependent on the species and environment present" - What is the evidence that oxalate has a negative effect on the gut microbiota? The authors clearly showed the negative effect of oxalate on the host. Although there are reports in the literature of oxalate consumers with a negative effect on the microbiome, such as Lactobacilli and Bifidobacteria, there is no evidence in this manuscript about a negative effect of oxalate on the microbiome, and there is not an experimental design to evaluate it.

These data are presented in Figure 2A and B. As stated, oxalate led to a net reduction in total microbial metabolites produced of 34 metabolites, with a significant shift in overall metabolome, indicative of metabolic inhibition. This is in comparison to the net gain of 9 metabolites, with no significant shift overall, in the mice with the NALB microbiota. The positive and negative effects of oxalate on the whole gut microbiota here are bolstered by previous studies on the effect of oxalate on pure cultures as discussed and cited on line 623624.

(6) Related to the last section, it is hard to really compare the results of the taxonomic cohort versus the metabolic cohort when the data of one cohort is in the main figure and the other in a supplementary figure. In addition, all the comparisons between the two cohorts seem to be qualitative. For any comparisons, the authors need to do a statistical comparison between the groups of the two cohorts.

The comparison of the two sets of data are indeed qualitative. This is because these mouse models were run in separate experiments to test separate hypotheses (whether utilization of specific substrates is enough to improve oxalate metabolism or if specific taxa previously responsive to dietary oxalate was better, which is stated in the manuscript). Given that these experimental models were tested separately, it would not be statistically valid to do a direct statistical comparison, even though the experimental procedures were the same and the only difference were the transplanted bacteria. The separation of the experiments into a main and supplemental figure was done out of necessity given the very large amount of data and many experimental mouse models that were run in this study overall.

Minor Comments.(1) The authors should define 'antinutrients'. This term is not a familiar concept and could create confusion.

This is defined in line 104 “molecules produced in plants to deter herbivory, disrupt homeostasis by targeting the function of the microbiome, host, or both”

(2) The authors should explicitly describe the N. albigulia, aka White-throated woodrat system, as early as possible in the result section.

We added some statements about the Swiss webster and N. albigula gut microbiota as poor and effective oxalate degraders in the second section of the results.

(3) SW-SW mice exhibited an oxalate-dependent alteration of 219 hepatic genes, with a net increase in activity. In comparison, the SW-NALB mice exhibited an oxalate-dependent alteration of 21 genes with a net decrease in activity. However, the visual representation of the PCoA in Figure 1B showed that the most different samples are the SW-NALB 0% and 1.5%. Could you please explain this difference?

In Figure 1b, the SW-NALB data are represented by the blue and black data points, which directly overlap with each other. The SW-SW data are the orange and purple data points, which exhibit very little overlap.

(4) Is Table S7 the same as Table S6? If not, there is a missing supplementary table.

These tables are different. We ensured that both are present.

(5) How did the authors test bacterial growth in in vivo studies (Figure 5B)?

We added a statement to the culturomic section of the methods – we used media with or without oxalate and quantified colony-forming units.

(6) A section of 16S rRNA metagenomics in the material and method section is not used across the main manuscript.

These data are presented in figures S7 and S10, as stated in the results. We added statements in the results to clarify that these figures show the 16S sequencing data.

(7) Lines 506-511: "Collectively, data from the current and previous studies on the effect of oxalate exposure on the gut microbiota support the hypothesis that the gut microbiota serves as an adaptive organ in which specific, metabolically redundant microbes respond to and eliminate dietary components, for the benefit of themselves, but which can residually protect or harm host health depending on the dietary molecules and gut microbiota composition." - What is the benefit to bacteria in eliminating oxalate? This is highly speculative to this system.

The benefit to bacteria is stated earlier in that paragraph – “In the current (Figs. 2B, 5B) and previous studies(33,34,64,65), we and others have found that oxalate can differentially exhibit positive or negative effects on microbial growth and metabolism dependent on the species and environment present.”

(8) Lines 504 -506: "Importantly, the near-universal presence of formate metabolism genes suggest that formate may be an even greater source of ecological pressure (Figures S2-S5)."- Formate is primarily produced by fermentative anaerobic bacteria, such as Bacteroides, Clostridia, and certain species of *Escherichia coli*, since formate would be present in anaerobic communities independently of oxalate. How is formate an even greater source of ecological pressure?

We added a statement about the toxicity of formate to both bacteria and mammalian hosts.